# Dynamic perceptual feature selectivity in primary somatosensory cortex upon reversal learning

Ronan Chéreau[1], Tanika Bawa[1,2], Leon Fodoulian[1,2], Alan Carleton [1], Stéphane Pagès[1] & Anthony Holtmaat [1✉]

Neurons in primary sensory cortex encode a variety of stimulus features upon perceptual learning. However, it is unclear whether the acquired stimulus selectivity remains stable when the same input is perceived in a different context. Here, we monitor the activity of individual neurons in the mouse primary somatosensory cortex during reward-based texture discrimination. We track their stimulus selectivity before and after changing reward contingencies, which allows us to identify various classes of neurons. We find neurons that stably represented a texture or the upcoming behavioral choice, but the majority is dynamic. Among those, a subpopulation of neurons regains texture selectivity contingent on the associated reward value. These value-sensitive neurons forecast the onset of learning by displaying a distinct and transient increase in activity, depending on past behavioral experience. Thus, stimulus selectivity of excitatory neurons during perceptual learning is dynamic and largely relies on behavioral contingencies, even in primary sensory cortex.

[1] Department of Basic Neurosciences and the Center for Neuroscience, CMU, University of Geneva, Rue Michel Servet 1, 1211 Geneva, Switzerland. [2] Lemanic Neuroscience Doctoral School, University of Geneva, Geneva, Switzerland. ✉email: anthony.holtmaat@unige.ch

The mammalian cortex encodes a myriad of sensory signal characteristics which are represented by neuronal assemblies, each with a preference for specific stimulus parameters[1,2]. It is believed that these assemblies are organized in a hierarchical fashion. First-order sensory areas encode lower-order stimulus features, such as texture coarseness[3–5], object orientation and direction[6], and sound frequency[7], whereas more complex features and contextual aspects of a stimulus are encoded by higher-order cortices[8–11]. Nonetheless, the coding in primary sensory cortices can exhibit higher levels of complexity, expressing non-sensory-related signals such as attention[12], anticipation[13], and behavioral choice[11–15]. Reward-based perceptual learning initially shapes the stimulus selectivity and response properties of primary sensory neurons, which may contribute to a reliable detection of particular features, and thereby improve perception[13,16–18]. However, it is unclear as to whether the stimulus preference of those neurons remains stable when the reward contingencies are changed. To study this, we monitor the shaping of stimulus selectivity for primary somatosensory cortical (S1) layer 2/3 (L2/3) neurons in mice that learn to discriminate between a rewarded and non-rewarded texture. We then reassess their selectivity upon reversal learning, which reveals a substantial subset of neurons that dynamically represents textures. Many lose or gain selectivity. Yet another class, which we term value-sensitive neurons, first lose and then regain texture selectivity contingent on the associated reward. The ramping up of this selectivity forecasts the onset of learning.

## Results

### Texture selectivity of L2/3 neurons increases with learning.

We trained mice on a head-fixed 'Go/No-go' texture discrimination task, similar to previous designs[5] (Fig. 1a). Thirsted animals were incited to lick a spout during a 2-s texture presentation in the form of a piece of P120 sandpaper (125-µm grit size; rewarded texture), in order to trigger the supply of a water reward at the end of the presentation period (scored as a 'hit' trial; Fig. 1a and Supplementary Fig. 1). A failure to lick was scored as a 'miss' trial. The animals needed to withhold from licking upon presentation of a P280 sandpaper (52-µm grit size; non-rewarded texture) to avoid a 200-ms white noise and a 5-s timeout period (scored as a 'correct reject' trial). A failure to withhold from licking was scored as a 'false alarm' trial (Fig. 1a and Supplementary Fig. 1). Mice learned to discriminate between the two stimuli (Fig. 1b). They typically started at chance level (naïve) and reached an average performance level of 82% within 3–7 days (expert mice) (Fig. 1b)[13–15,19]. To verify that the task was whisker-dependent and involved the cortex, we trimmed the whiskers ipsilateral to the texture, or suppressed contralateral cortical activity using a local injection of the γ-aminobutyric acid receptor (GABAR) agonist muscimol in separate sets of expert mice (see Methods). Both treatments reduced the performance to chance level (Fig. 1c, d). This indicates that to solve this task mice fully rely on somatosensory input and do not use additional sensory information, and that the task involves signal processing through S1.

In order to monitor the activity of S1 neurons during texture discrimination learning, we co-expressed the genetically encoded calcium sensor GCaMP6s and the cell filler mRuby2, predominantly in excitatory L2/3 neurons using adeno-associated viral vectors (Fig. 1e, Supplementary Fig. 2)[20]. Single-cell calcium signals were recorded using two-photon laser scanning microscopy (2PLSM; Fig. 1e, f). Fast-volumetric imaging was performed to allow for the correction of axial motion artifacts (Fig. 1e, Methods section)[21].

Similar to previous studies[5,14], a fraction of the neurons displayed a differential response to the textures (Fig. 1f and Supplementary Fig. 3). In order to determine the texture selectivity of individual neurons during learning we compared the calcium signal amplitudes evoked by the two different sandpapers using a receiver-operating characteristic (ROC) curve analysis. This provided a discrimination index for each neuron (DI; Fig. 1g, Methods section)[22]. On average, the fraction of selective neurons increased with learning (Fig. 1h). Interestingly, we observed that in expert mice, a larger fraction of the recorded population was selective for the P120 (rewarded) as compared with the P280 (non-rewarded) texture (Fig. 1i) and that this difference built up with learning (Fig. 1j).

What could be the cause of the increase in selectivity bias during learning? One explanation holds that the neuronal responses strictly correlate with the different behaviors the animals exhibit during Go and No-go trials, which emerges with learning (Supplementary Fig. 1). In that case, the neuronal activity could be linked to the motor-output that is associated with licking, and not exclusively to the presented texture. Alternatively, L2/3 neurons could encode higher-order features that are associated with the textures (such as the reward value or the behavioral choice). To explore these possibilities, we first conducted experiments that allowed us to categorize neurons based on their activity in relation to the animal's licking and whisking behavior, and then we reassessed their selectivity after inverting the reward-contingencies.

### Neuronal activity represents sensory input.

We first investigated the possibility that the P120-selective neurons were merely reporting licking, by comparing for all hit trials, the delay between the onset of the calcium signal and the time of texture presentation or the time of the 1st lick. For the majority of neurons, the rise in the calcium signal occurred immediately after the texture presentation and preceded the 1st lick with a larger jitter (Fig. 2a–c). This suggests that the activity of the P120-selective neurons was evoked by the texture and not by licking. However, this analysis did not exclude the possibility that selectivity had been influenced by an increasingly stereotyped behavioral sequence during learning, including whisking. To dissociate sensory-evoked neuronal activity from activity that was primarily related to whisking or licking we exposed mice to the various task-related stimuli before the training had started. The stimuli were presented separately and without a temporal structure (Fig. 3a). We also monitored the animal's whisking and licking behavior. Together, this allowed us to categorize neurons based on their activity in relation to the sound cue, texture presentation, as well as whisking and licking behavior. We found that a large fraction of neurons (36.7% of the total population) exhibited touch-related activity during texture presentation while few neurons were sensitive to the auditory cue (0.8%; Fig. 3b). Within the pool of touch-sensitive neurons there was no bias in texture selectivity (Fig. 3c). This suggests that the imaged population was not a priori preferring any of the two textures, which is in line with previous work[4]. Then we determined whether neurons showed whisking or licking-related responses. We trained a random forests machine-learning model using the inferred firing rates from the calcium signal to assess for each neuron if its activity could predict whisking and/or licking rates. The model was trained using a range of positive and negative time lags of the neuronal activity relative to behavior, in order to account for possible pre-motor related activity (i.e. preceding the behavior) and/or sensory-related activity (i.e. following the behavior). For each neuron we calculated the prediction power (PP), which reflected the correlation between the animals' actual whisking and licking behavior, and the behavior that was predicted by its activity (Fig. 3d). We plotted the PP distributions for whisking

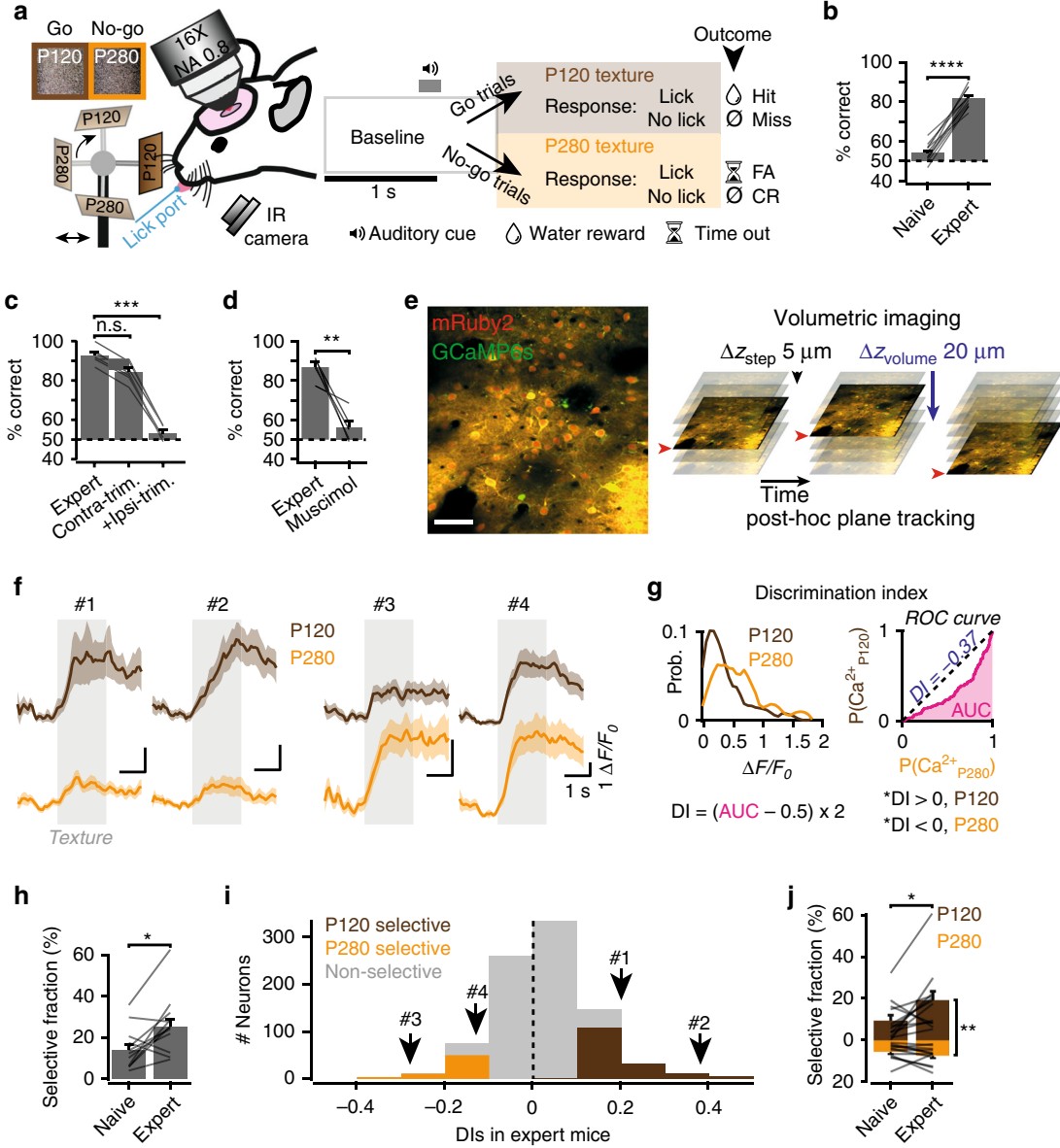

**Fig. 1 Increase in texture selectivity of S1 neurons during discrimination learning. a** Experimental setup and structure. Mice were rewarded for licking the water spout upon random presentations of the P120 but not the P280 sandpaper. **b** Discrimination performance across learning ($N = 12$ mice, Wilcoxon rank-sum test, ****$P = 3.7 \times 10^{-5}$). **c** Effect of contra and ipsilateral whisker trimming on behavioral performance ($N = 7$ mice, Friedman test and post-hoc Dunn's test, n.s. $P = 0.33$, ***$P = 0.0009$). **d** An injection of muscimol in S1 of expert animals reduced behavioral performance ($N = 5$ mice, Wilcoxon rank-sum test, **$P = 0.0079$). **e** Left, example of field of view containing mRuby2/GCaMP6s-expressing neurons (scale bar: 100 µm). Right, thin volumetric imaging was performed in order to correct post-hoc for axial brain motion artefacts. **f** Examples of average calcium signal to the P120 texture (brown) and P280 texture (orange). **g** Example of ROC analysis for a neuron during an expert session. Left, probability distributions of the calcium signals for the P120 and P280 trials. Right, ROC curve from which the DI is calculated. The DI expresses the likelihood that the neuronal calcium signals predicted the presented texture. Statistical significance was determined using a permutation test (see Methods). DIs for example neurons #1-4 in **f** are 0.20, 0.38, −0.28, and −0.13, respectively ($P < 0.05$ for all DIs). **h** The fraction of selective neurons before and after learning ($N = 12$ mice, 875 neurons, Wilcoxon rank-sum test, *$P = 0.03$). **i** Distribution of DIs in expert mice ($N = 12$ mice) with example neurons from **f**. P120-selective neurons (brown): 146/875; P280-selective neurons (orange): 57/875. **j** The fraction of P120 (brown) and P280 (orange) selective neurons before and after learning ($N = 12$ mice, 875 neurons, Wilcoxon rank-sum tests, P120-selective fraction in naïve vs expert, *$P = 0.02$; P280-selective fraction in naïve vs expert, $P = 0.33$; P120 vs P280 selective fractions in naïve, $P = 0.36$, and expert, **$P = 0.006$). The bars with error bars, or traces with shaded areas represent averages ±SEM. Lines between bars represent individual mice.

and licking rates as inferred from the GCaMP6s signal. This was compared to a control distribution that was inferred from the mRuby2 signal to assess the noise in PP measurement (Fig. 3e, f). Neurons with a PP over a threshold criterion of five standard deviations above the mean of the control distribution were considered to be predictive of whisking and/or licking. We found that 9.4% of the neurons were partially predicting the animal's whisking rate whereas only 2% predicted licking rates (Fig. 3e–g). We then compared the resulting categories with the selectivity that the neurons displayed in the subsequent texture

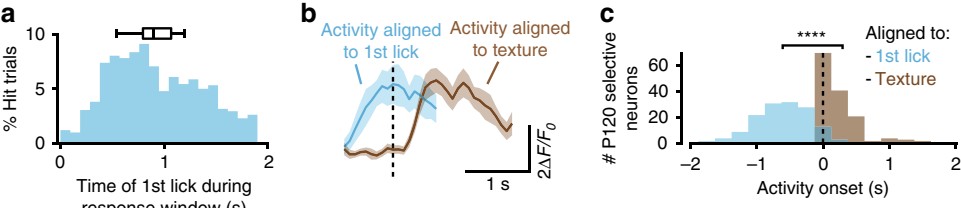

**Fig. 2 Activity of P120 selective neurons coincides with texture presentation. a** Distribution of reaction times over all hit trials across expert mice ($N =$ 923 trials). Box plot shows mean, 25th–75th percentiles, and the minimum and maximum average reaction times over mice ($N = 12$ mice). **b** Example of the average calcium signal in a P120 selective neuron of an expert mouse, aligned to texture presentation (brown) and the 1st lick (blue). Shaded areas represent SEM. **c** Distribution of the response onsets for all P120 selective neurons in expert sessions aligned to the texture presentation (brown) or to the 1st lick (blue, $N = 12$ mice, 146 neurons, paired $t$-test, ****$P = 2 \times 10^{-46}$).

discrimination task. Most of the neurons that were found to be selective after training had formerly been categorized as undefined or reporting touch (88%; Fig. 3g). Altogether, these data strongly suggest that the stimulus-selective neurons did not exclusively signal whisking or licking behavior during the task. Moreover, only 11% of the P120-selective neurons were predicting the animal's whisking rate and 0% the licking rate. Thus, the biased increase in P120 selectivity during texture discrimination learning could not be explained by mere changes in the animal's whisking or licking behavior.

**Texture selectivity is dynamic upon reversal learning.** Studies using comparable paradigms have reported that S1 neurons exhibit selectivity not only for the tactile stimulus but also for the behavioral choice[5,14,15]. In order to test this, we uncoupled the behavioral choice from the respective textures by inverting the reward contingencies. This allowed us to assess which neurons were persistently selective for a given texture, and which were dynamic. To this end, expert mice were continued to be trained on the same textures, but now the detection of the P280 texture was rewarded and the P120 texture was not (Fig. 4a). Upon reversal the performance initially dropped to chance level (the post-reversal naïve phase; Fig. 4b) before it reached the expert criterion again within 2–4 days (the post-reversal expert phase). In the post-reversal naïve phase, the neuronal population's average DI remained of the same sign as compared with the pre-reversal expert phase. However, we observed an inversion of the DI's sign in the post-reversal expert phase (Fig. 4c), indicating that many neurons had changed their texture selectivity during reversal learning. By comparing the DI of each neuron over expert sessions before and after reversal we could define a variety of neuronal classes, including those that remained selective for the same texture (4%; e.g. neuron 1 in Fig. 4d), those that reversed their selectivity to the other texture (and thus invariably reported textures contingent on the associated reward, 8%; e.g. neuron 2 in Fig. 4d), and those that had lost (19%) or gained (18%) selectivity altogether (Fig. 5a–c). Overall, the population regained a selectivity bias for the rewarded texture (Fig. 5c, d). The changes in selectivity could be the result of network plasticity. To assess this, we calculated the level of co-fluctuation in spontaneous activity within the groups that had lost or gained selectivity, which may reflect the level of mutual connectivity[23–25]. Upon reversal, the level of co-fluctuation increased for gained neurons and decreased for lost neurons (Fig. 5e). This may indicate that reversal learning promotes the rewiring of local synaptic circuits.

We also checked whether the various classes correlated with the animal's whisking or licking behavior. We found no difference in the average calcium signal for any of the classes above when comparing trials for which the animal displayed high whisking or licking rates with low-rate trials (Fig. 5f, g and Supplementary Fig. 4a, b). This result is in line with the decoding model (Fig. 3) and indicates that the dynamics in selectivity observed after reversal learning cannot be attributed to alterations in whisking and licking.

Altogether, the reversal learning experiment shows that texture selectivity of L2/3 neurons in S1 is largely dynamic, with a fraction of neurons reversing their texture selectivity congruent with the reward contingency. This suggests that although for some neurons selectivity is determined solely by the texture attributes of the stimuli, for many others it is shaped by higher-order features that are associated with the stimuli.

**Selectivity reversal is associated with choice or reward.** What determines the selectivity dynamics in the class of neurons that followed the textures' reward contingencies? We envisioned two possibilities. Neurons could persistently report the upcoming choice[5,14,15], independent of reversal learning. Alternatively, neurons could gradually update their texture selectivity during reversal learning, congruous with the associated reward. The latter neurons would therefore signal the texture value rather than upcoming behavioral choice, as seen in other brain areas[8,9,26,27]. To address this, we tracked the responses of the reversibly selective neurons according to the trial outcome (hits, misses, FAs, and CRs) throughout the reversal learning process. We distinguished three learning phases: pre-reversal expert, post-reversal naïve, and post-reversal expert. Upon reversal of the reward contingencies, some neurons showed persistently larger responses during hit and FA trials as compared with miss and CR trials (e.g. Neuron 1 in Fig. 6a; Supplementary Fig. 5). Other cells exhibited larger responses in hit and miss trials during the pre-reversal expert phase, then showed larger responses in FA and CR trials during the naïve post-reversal phase, and finally regained response strength in hit and miss trials during the expert post-reversal phase (e.g. Neuron 2 in Fig. 6a; Supplementary Fig. 5). Thus, whereas the former neuron stably preferred a texture congruent with the final action-selection (i.e. choice) throughout all phases, the latter neuron updated its selectivity during re-learning, possibly based on the reward-outcome that was associated with the texture (i.e. value). The difference between those two neurons became most striking during the post-reversal naïve phase in which the animals typically abandoned their previous behavioral strategy and made inconsistent choices. This allowed us to parse out from the class of reversibly selective neurons those whose selectivity was conforming to the animal's upcoming choice to lick or not to lick (i.e. choice neurons) or conforming to the texture's associated reward value (i.e. value neurons). To quantitatively parse the different types of selectivity, we calculated a choice index (CI) for each neuron. Similar to the DI, this was

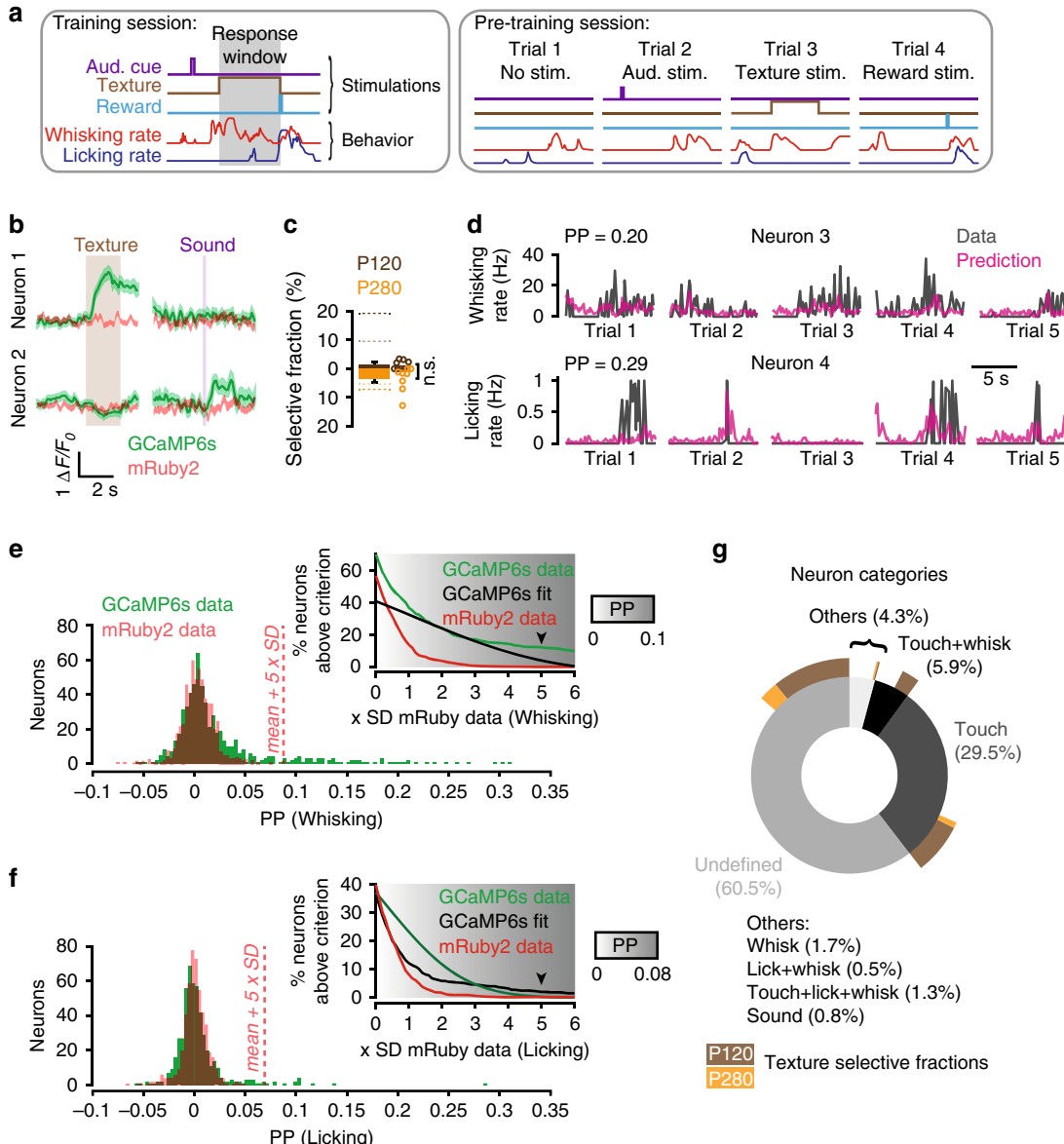

**Fig. 3 Activity of P120 selective neurons is not predicting licking behavior. a** Examples of training and pre-training trials with whisking and licking rates. Left, structure of a training trial. Right, pre-training trials in which the 2-s texture presentation, auditory cue, or water reward was delivered independently and in a pseudo-random fashion. **b** GCaMP6s and mRuby2 traces of two neurons that significantly responded to texture (neuron 1, touch neuron) or sound (neuron 2). Shaded areas represent SEM. **c** The fractions of touch neurons that are selective for P120 (brown) or P280 (orange) pre-training ($N = 8$ mice, 549 neurons, Wilcoxon rank-sum test, $P = 0.27$). Dashed lines indicate the fraction of neurons that became selective during training in naïve (thin) and expert (thick) phases (from Fig. 1j). Open circles represent individual mice. **d** Examples of two neurons decoding whisking or licking rates as predicted by a Random-Forest decoder that was trained on inferred neuronal spike rates (Methods section), and whisking and licking rates. Actual rates (Data) are superimposed with the scaled model prediction (Prediction). PP prediction power. **e**, **f** PP distributions for whisking (**e**) and licking (**f**) rates reflecting the similarity between the behavioral data and the random forest decoding model based on the activity of individual neurons. The signal intensity fluctuation recorded from the mRuby2 channel was used as control. A neuron with a PP above 5 standard deviations (SD) of the control distribution was considered significantly predictive of the behavioral feature ($N = 591$ neurons, 55 neurons predicted whisking rate and 10 neurons predicted licking behavior). The inset shows the percentage of neurons above a variable threshold criterion from the measured GCaMP6s PP values (light green) compared with normally distributed GCaMP6s PP values (dark green) and the measured mRuby2 PP values (red). The fraction of neurons detected as predictive for whisking and licking is higher than chance for criterion values above 5 SD (arrowhead) of the mRuby2 distribution. **g** Percentage of neurons in the various categories as determined by combining the response (**b**) and decoding (**e**, **f**) analysis ($N = 505$ neurons, 8 mice). The outer ring represents the fraction of selective neurons in the expert phase belonging to each category.

based on a ROC curve analysis, but now comparing the response amplitudes between lick and no-lick trials (Fig. 6b). This analysis confirmed the existence of the two subclasses (Fig. 6c), one for which the CI remained stable throughout the naïve phase after

reversal (choice neurons), and one for which the CI was altered (value neurons). For both classes, the calcium signals did not correlate with the whisking and licking rates (Fig. 6d and Supplementary Fig. 4c, d). In addition, only a few neurons in both

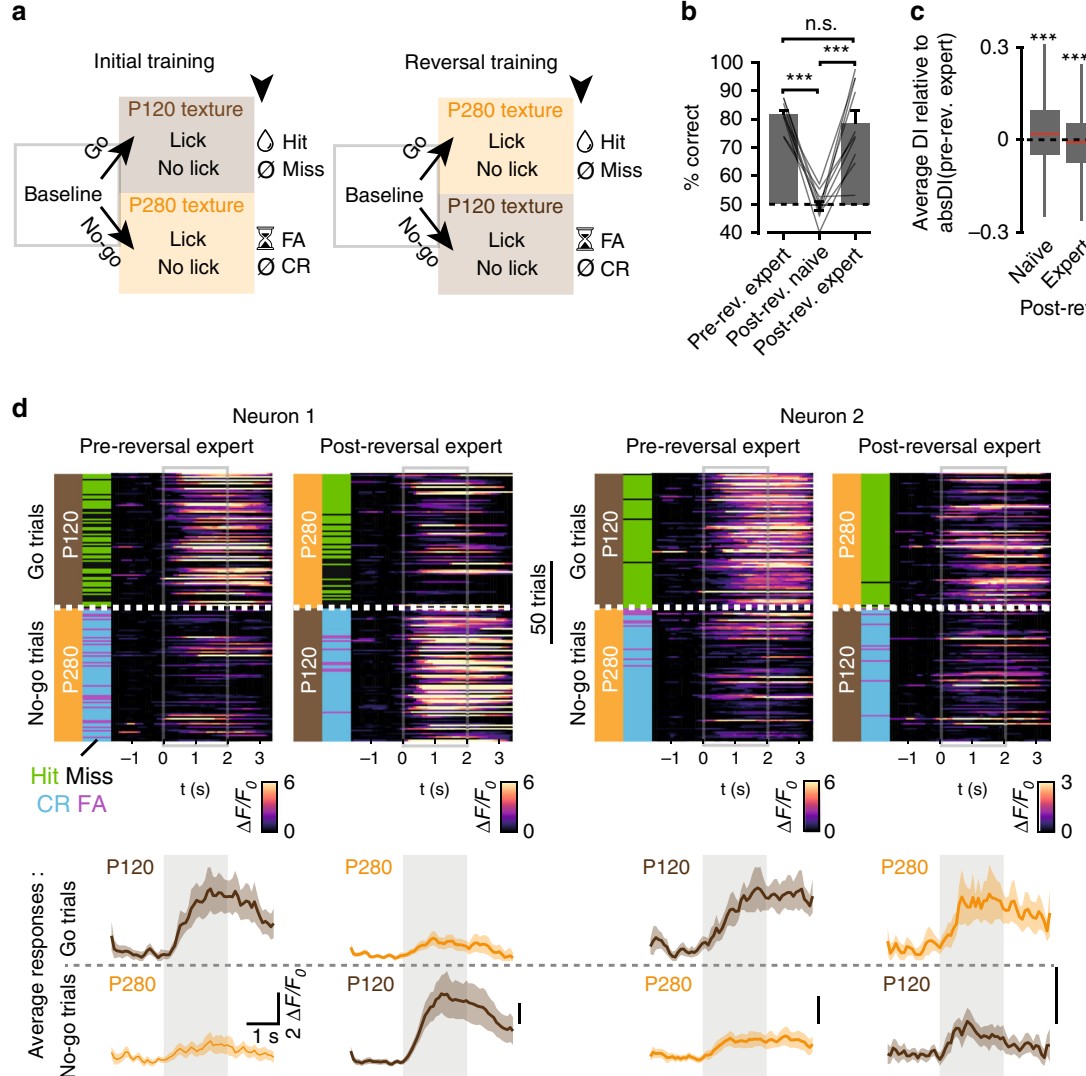

**Fig. 4 Evolution of neuronal selectivity during texture reversal learning. a** Trial structure for the initial and reversal training. **b** Average performance of mice in pre-reversal expert, post-reversal naïve, and post-reversal expert sessions ($N = 12$ mice, rank-sum test, ***$P = 1.8 \times 10^{-4}$, n.s. $P = 0.62$). Error bars represent SEM, and lines individual mice. **c** DI in post-reversal naïve and expert phases relative to their absolute pre-reversal expert DI. The box plots show the median (red line), interquartile (box), and range (whiskers) of the data. Outliers are not represented. In the post-reversal naïve phase, the selectivity of the population remains of the same sign, indicating that neurons still preferentially respond to the previously rewarded P120 texture. When mice become experts again, the selectivity of the population reverses to the newly rewarded P280 texture ($N = 255$, one sample $t$-test ***$P = 9 \times 10^{-4}$, ****$P = 2.9 \times 10^{-13}$). **d** Single trial responses of two example neurons from expert sessions pre and post texture reversal aligned to texture presentation and sorted by stimulus type (160 trials from these expert sessions are displayed). Corresponding trial outcomes and performance are also shown. The neuron on the left remains selective for the P120 texture whereas the neuron on the right reversed its selectivity to the P280 texture. Below, average calcium signals for these trials and their DIs. Shaded regions represent SEM.

classes had previously been categorized as being predictive for whisking + licking, similar to the other classes of neurons (Fig. 6e). This confirms that the selectivity dynamics (or lack thereof) in choice and value neurons could not be attributed to alterations in whisking or licking.

Altogether, this shows that reversibly selective neurons could be sub-divided into two classes: neurons that signaled the stimulus congruent with the animal's upcoming choice and neurons that reported the contextual stimulus value (Fig. 7a). To illustrate the differences between these classes, we provide examples of the temporal evolution of the DI and CI throughout reversal learning for a choice neuron and a value neuron from the same animal (Fig. 7b). In line with our previous analysis (Fig. 6c), the DI of both neurons showed a relatively similar temporal

profile, with an initial drop after reversal and a gradual inversion during re-learning. On the other hand, the CI of the choice neuron remained positive throughout the reversal learning phases, whereas the CI of the value neuron did not. Notably, for the value neuron the inversion of the DI seemingly occurred tens of trials before the animal's performance started to increase, whereas for the choice neuron the inversion coincided sharply with the increase in performance.

**Value neurons display error history activity during learning.** Based on the preceding observations, we hypothesized that the gradual reacquisition of texture preference by the value neurons carries a signal that predicts the upcoming improvement in the

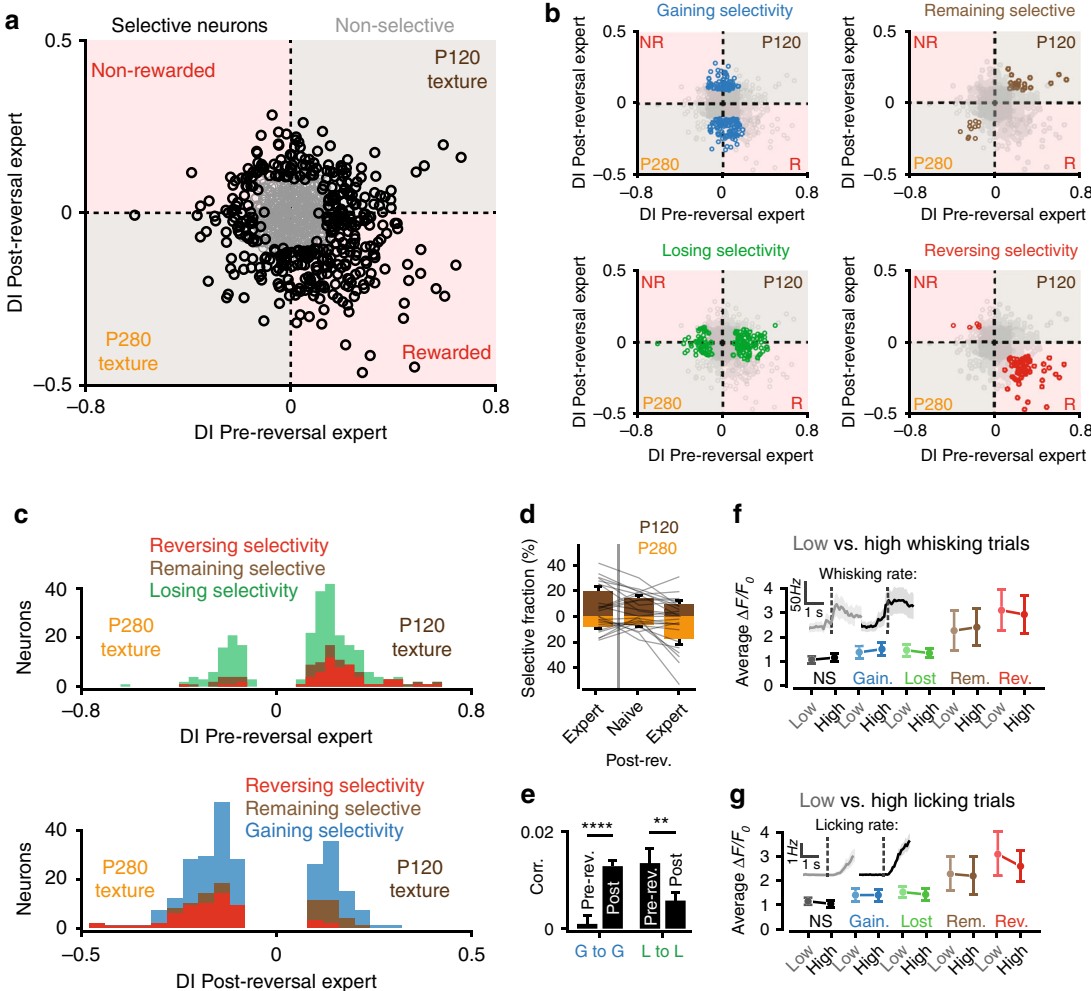

**Fig. 5 Classification of neuronal selectivity upon texture reversal learning. a** DIs of neurons in expert sessions pre vs. post-reversal. Neurons that were selective in at least one of the two learning phases are in black ($N = 388$, 12 mice) and non-selective neurons ($N = 412$) in gray. The upper-right and lower-left quadrants represent selectivity for the P120 and P280 texture, respectively. The lower-right and upper-left quadrants represent selectivity for the rewarded and non-rewarded texture, respectively. **b** From the pool of selective neurons in **a**, four classes could be extracted by comparing their selectivity pre vs. post-reversal: gaining selectivity (blue, $N = 144$), non-selective pre-reversal but selective post-reversal. Losing selectivity (green, $N = 152$), selective pre-reversal and non-selective post-reversal. Remaining selective (brown, $N = 33$), selective for the same texture pre and post-reversal. Reversing selectivity (red, $N = 59$), selective for the opposite texture pre and post-reversal. NR non-rewarded, R rewarded. **c** Distributions of DIs of selective neurons pre (top) and post-reversal (bottom). **d** Average fractions of P120 and P280-selective neurons across mice pre and post-reversal ($N = 12$ mice, 800 neurons). Lines represent individual mice. **e** Correlations of average spontaneous between pairs of neurons that gained (G to G) or lost selectivity (L to L) comparing pre-reversal expert to post-reversal expert (G to G: 224 pairs, L to L: 360 pairs, paired $t$-tests, ****$P = 6 \times 10^{-9}$, **$P = 0.0014$). **f** Average calcium signals upon texture presentation for each neuronal class when separated by hit trials with low (light) and high (dark) whisking rates. Inset, average whisking rates across low (blue) and high (red) rate trials (for classification, Methods section). Calcium signals were not significantly different between low and high whisking rates (Paired $t$-test, Non-selective (NS): $N = 412$ neurons, $P = 0.66$, Gaining selectivity (Gain): $N = 144$ neurons, $P = 0.99$, Losing selectivity (Lost): $N = 152$ neurons, $P = 0.80$, Remaining selective (Rem): $N = 33$ neurons, $P = 0.82$, Reversing selectivity (Rev): $N = 59$ neurons, $P = 0.86$, across 12 mice). **g** Same analysis as in **f** but for licking rates, showing no difference between low and high rates (Paired $t$-test, NS: $N = 412$ neurons, $P = 0.90$, Gain: $N = 144$ neurons, $P = 0.98$, Lost: $N = 152$ neurons, $P = 0.68$, Rem: $N = 33$ neurons, $P = 0.98$, Rev: $N = 59$ neurons, $P = 0.62$, across 12 mice).

animal's texture discrimination performance. Such a signal might consist of distinct response amplitudes during certain trials, which could depend on whether the animal had previously made correct or incorrect choices[26,28–30]. Previous work suggests that a correct trial that follows an incorrect trial is considered more instructive for the animal than two consecutive correct trials[8,9,26]. To test this, we focused our analysis on those consecutive trials in which mice were actively licking upon texture presentation (i.e. hits and FAs), hence ensuring that they were engaged in the task. We compared the mean response amplitudes of hit trials that were preceded by a FA trial ($R_{\text{hit(post FA)}}$) to those that were

preceded by a hit trial ($R_{\text{hit(post hit)}}$) (Fig. 8a). All trials across mice were aligned to the point at which the reversal learning had reached the expert criterion (Fig. 8b, Supplementary Fig. 6, and Supplementary Table 1). Averaged hit and FA rates over a 200-trial rolling window separated from one another at ~140 trials before the expert criterion. This point indicated the moment at which mice started to improve their performance, which we defined as the learning onset (Fig. 8c, black arrow head). For non-selective neurons as well as choice, texture, gained, and lost selectivity neurons, we did not observe any difference between the $R_{\text{hit(post hit)}}$ amplitudes and the $R_{\text{hit(post FA)}}$ amplitudes (Fig. 8d).

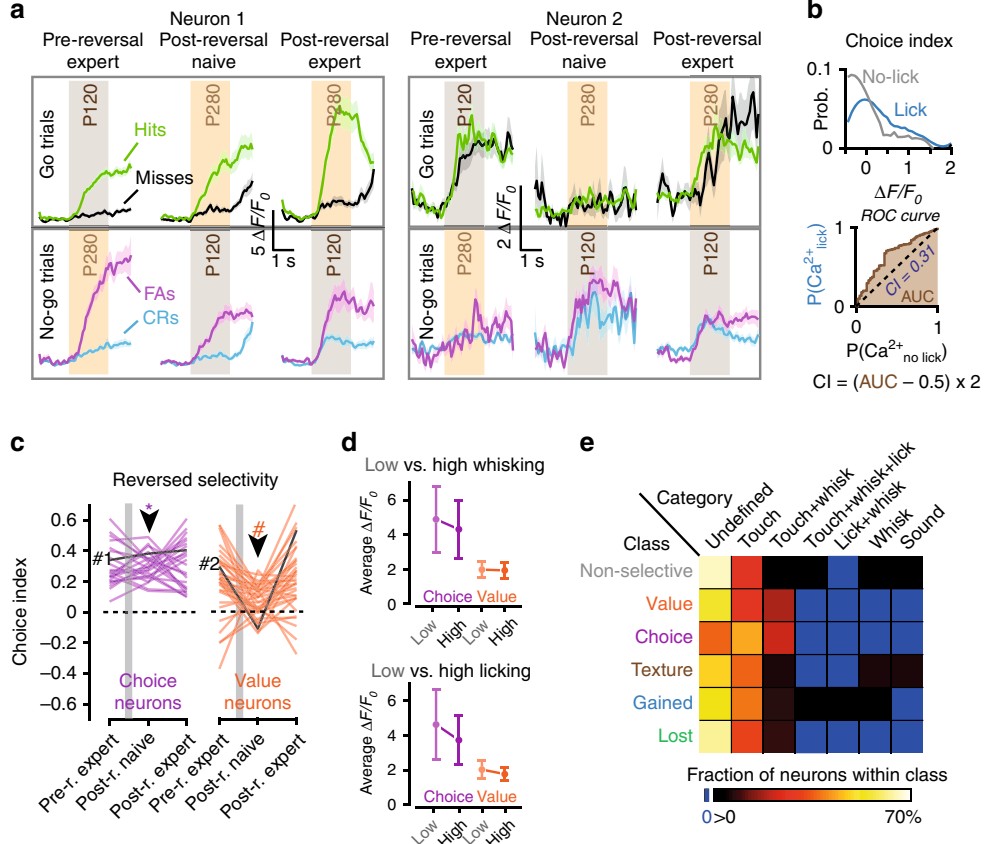

**Fig. 6 Neurons reverse selectivity congruent with choice or reward-value. a** Average calcium signals in hits, misses, CRs, and FAs for two neurons that reversed selectivity during reversal learning. Neuron 1, responses remained high for Hit and FA trials throughout all phases. Neuron 2, responses decreased for hits and misses and increased for FA and CR trials during the naïve reversal phase, and then increased again for hits and misses in the reversal expert phase. **b** Example of the ROC analysis for calculating the choice index (CI) of a neuron. The calculation is similar to the DI except that calcium signals were sorted by the animal's licking behavior (lick vs. no-lick trials). Top, probability distributions of the calcium signals for the lick and no-lick trials. Bottom, ROC curve from which the CI is calculated. The CI expresses the likelihood that neuronal calcium signals predicted the upcoming lick. Statistical significance was determined using a permutation test (Methods section). **c** Sub-classification of neurons that reversed selectivity during reversal learning based on changes in choice index (CI). Choice neurons (purple, $N = 24$), CI was significant and remained of the same sign throughout all phases (*). Value neurons (orange, $N = 35$ neurons), CI was either not significant or of the opposite sign in post-reversal naïve sessions (#). **d** Average calcium signals upon texture presentation for each neuronal class when separated by hit trials with low (light) and high (dark) whisking or licking rates. Calcium signals were neither influenced by whisking rates (Paired $t$-tests; choice: $N = 24$, $P = 0.87$, value: $N = 35$, $P = 0.81$) nor licking rates (choice, $P = 0.67$, value, $P = 0.77$). Error bars represent SEM. **e** Table showing the percentages of each category within the different functional classes. Notably, none of the choice and value neurons was categorized as predictive for licking (number of neurons per class for which corresponding categories were defined prior training: non-selective 271, Value 11, Choice 14, Texture 17, Gained 90, and Lost 75). 0% is represented in blue in the look up table.

In contrast, for the contextual value neurons, the average $R_{hit(post FA)}$ response amplitudes became larger than the $R_{hit(post hit)}$ amplitudes, at ~260 trials before the expert criterion, and ~120 trials before learning onset (Fig. 8c, d, red arrowhead). The two types of responses became similar again when mice performed above the expert criterion. During this interval, we did not observe a change in the sampling strategy of the texture confirming that the difference in responses is not associated with changes in licking and/or whisking rates (Supplementary Fig. 7). We used the normalized difference between $R_{hit(post FA)}$ and $R_{hit(post hit)}$ responses as an error history index (Fig. 8e), and observed that a large fraction of the value neurons exhibited a transient increase in the error history as compared to the other neuronal classes that we had identified. Such an anticipation of the learning onset could not be deduced from the DI evolution (Supplementary Fig. 8). Altogether, these results indicate that the change in texture preference of value neurons caries a signal that is indicative of the upcoming improvement in discrimination, i.e. learning.

## Discussion

Previous studies indicate that reward-based perceptual learning increases the reliability and selectivity of neuronal responses in primary sensory cortices. As a consequence, the neuronal population that represents the relevant sensory stimuli stabilizes, which may improve perception[13,16–18,31]. We extend on this work by tracking the stimulus feature selectivity of neurons in mouse S1, first during learning of a Go/No-go texture discrimination task (Fig. 1), and subsequently upon reversal learning of the task (Figs. 4 and 5).

We found that during learning, the population of neurons displaying selectivity for the rewarded texture became increasingly larger than for the non-rewarded texture (Fig. 1h–j). This finding agrees with previous studies describing that response selectivity is shaped by the behavioral choice of the animal[5,11,14,15]. Using the reversal learning paradigm we then showed that whereas a small population of neurons can stably encode a texture, a large fraction loses, gains, or first loses and

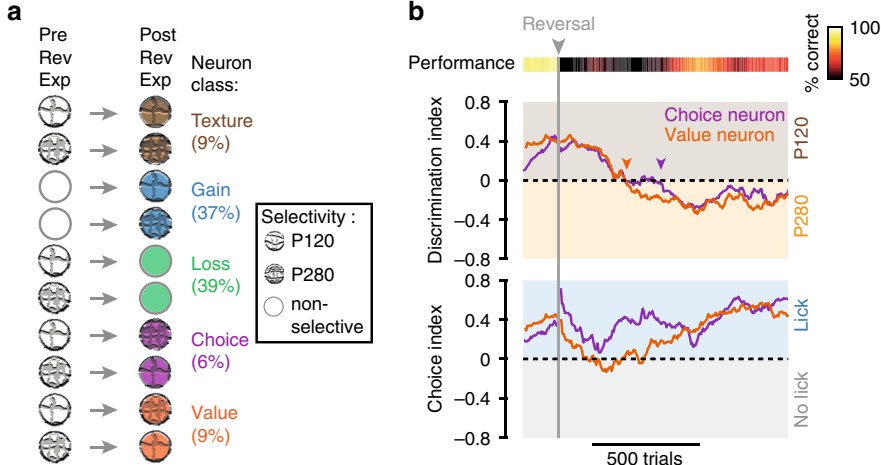

**Fig. 7 The temporal evolution of discrimination and choice indices during reversal learning. a** Summary of the different classes of neurons as revealed by reversal learning. Neurons either remained selective to the original texture (brown), gained (blue), or lost (green) selectivity for the stimulus, or reversed their selectivity. From the pool of neurons that were reversibly selective, some followed the animal's choice (purple) and others signaled the reward-associated stimulus value (orange). **b** The temporal profile of the texture (top) and choice discrimination (bottom) indices for a choice (purple) and a value neuron (orange), before and during reversal learning. Both neurons initially show preferential responses for the P120 texture, even after reversal, but then shift their response preference toward the P280 texture during relearning. For the choice neuron, the response preference remains high for the upcoming lick throughout training even in the reversal naïve phase. For the value neuron, the preference is unstable throughout relearning. Note that the preference of the value neuron inverted prior the increase in the mouse's performance whereas the preference inversion of the choice neuron seemed to follow the level of performance (orange and purple arrowheads, respectively).

then regains selectivity for a texture when the reward contingencies are reversed (Figs. 4 and 5). This implies that a simple alteration of reward contingencies can disorganize a pre-established selectivity map in S1, which is then extensively reshaped with relearning.

The reshaping of this map could be the result of plasticity mechanisms that also underlie the experience-dependent tuning of neuronal response properties in primary sensory cortex[20]. In this case, Hebbian plasticity may drive the phenomenon, with the result that similarly tuned neurons become more strongly interconnected[23–25]. This is supported by our finding that both, neurons that gain selectivity and those that lose selectivity show higher co-fluctuation in spontaneous activity during the time they are selective (Fig. 5e).

The reshaping of the selectivity map during reversal learning is remarkable, since the lower-order sensory features that are embodied in the textures had not changed. Thus, in principle, the capacity of the S1 neuronal population to discriminate those lower-order sensory features did not need to be modified in order for the mouse to resolve the altered reward contingencies. Nonetheless, the finding is congruent with the idea that learning continuously optimizes sensory representations in cortex, and that this strongly depends on the stimulus context[2,13,14]. In our study, the reward contingency could represent an important aspect of the context that modulates sensory representations. Indeed, the selectivity dynamics in the neuronal population upon reversal learning suggests that neurons in S1 do not solely represent lower-order sensory features. Instead, they seem to selectively report the association between a lower-order stimulus feature and a paired higher-order feature, such as the reward.

In Go/No-go tasks the reward is tightly coupled to the animal's choice for licking. Thus, the selectivity for the texture-reward coupling could merely represent the encoding of the upcoming behavioral choice. The reversal learning paradigm allowed us to assess the stability of the neuronal responses for this coupling, e.g. whether the initial P120-selective neurons stably respond to the animal's choice, even during the post-reversal naïve phase,

or whether they lose selectivity shortly upon reversal and then re-built it with relearning (Figs. 6 and 7). We found that more than half of the P120-selective neurons belonged to the latter class. Thus, their sensory responses were transiently uncoupled from the animal's choice, and primarily depended on whether the presented texture was associated with the upcoming reward (or not), i.e. the value of the texture. In future experiments it will be interesting to test whether repeated reversal learning continues to renew the selective population, or whether the population reverts back to the original response configuration.

Previous studies indicate that the value of a sensory stimulus is encoded by higher-order areas such as the posterior parietal, orbitofrontal, and retrosplenial cortices[8,9,26,27]. Our data shows that value-encoding is also an attribute of a population of neurons in S1. The instructive cues for this selectivity could be manifold. For example, they could be provided by direct feedback from the aforementioned higher-order cortical areas, or they could be derived from sub-cortical areas that are implicated in attention and behavioral updating during learning[32,33]. Modulatory reinforcement signals that are associated with behavioral outcome could also play a major role[33–35]. Indeed, reward-related response modulation has been observed in S1[28], and was found to promote cortical plasticity processes related to visual response tuning in primary visual cortex[16]. We found that the value neurons gradually regained their preference for the rewarded texture with relearning, which would be congruent with the idea that reward-related plasticity mechanisms contribute to shaping perceptual representations in cortex.

At this point it is not clear if the value neurons constitute a specific subpopulation of L2/3 neurons. Since we used an AAV expression cassette with a generic promoter, the population of value neurons could theoretically contain interneurons. L2/3 of S1 contains various types of interneurons of which vasoactive intestinal peptide (VIP)-positive interneurons have been shown to be implicated in shaping neuronal responses[35–38] and cortical plasticity[39,40]. It is tempting to speculate that the reward-related response modulation that we observed is conveyed by VIP interneurons[33,41].

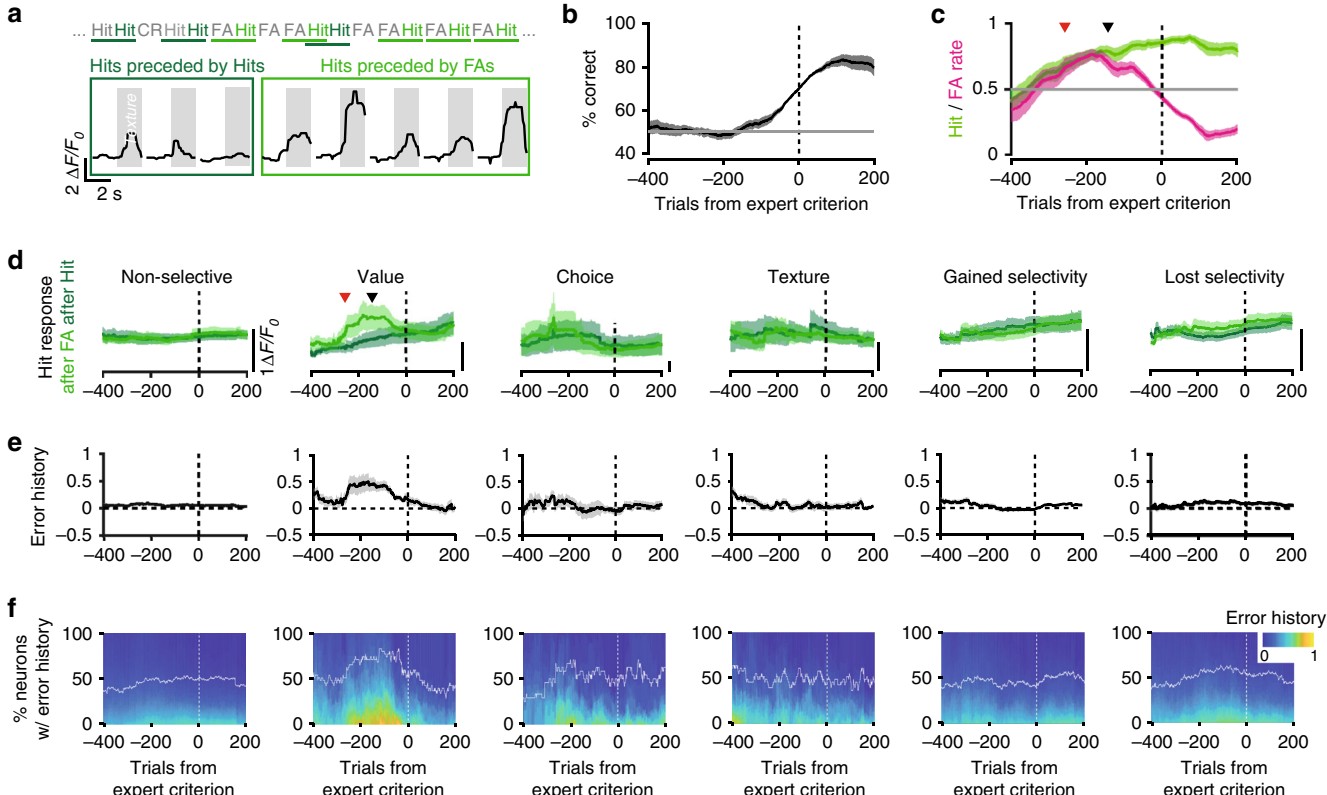

**Fig. 8 Behavioral error history is signaled by value neurons during reversal learning prior to performance increase. a** Responses of a value neuron in hit trials during reversal learning but before the behavioral performance increases. Hit trials preceded by a false alarm trial (FA, light green) have larger responses than hit trials that were preceded by another hit trial (dark green). **b** Average performance curves across mice realigned to the expert criterion (70% correct, dotted line) post-reversal, calculated with a rolling window of 200 trials ($N = 10$ mice containing value neurons; Supplementary Table 1). **c** The corresponding average hit and FA rate averages. **d** The corresponding average calcium signals ($\Delta F/F_0$) in hit trials that followed upon another hit trial (dark green) or upon a FA trial (light green) for the various neuronal classes: non-selective ($N = 412$, 12 mice), value ($N = 35$, 10 mice), choice ($N = 24$, 6 mice), texture selective ($N = 33$, 7 mice), gained ($N = 144$, 12 mice), and lost selectivity ($N = 152$, 12 mice). The arrowheads in **c** and **d** indicate the trials at which the 95% confidence intervals separate for hit and FA rates (black), and for the responses in the hit–hit vs. the FA-hit trials (red). **e** Calculated mean error history representing the normalized difference between the two types of hit responses (Methods section). **f** Percentage of neurons per class with significant error history rates (white line) above the cumulative distributions of error history values. Shaded areas in **b–e** represent 95% confidence intervals.

We also found that value neurons transiently displayed enhanced response amplitudes dependent on the animal's behavioral error history (Fig. 8). During the naïve reversal phase these neurons showed higher responses in hit trials if the hit trial was preceded by a false alarm trial. This phenomenon was prominent during the transition from the naïve to expert reversal phase and forecasted the increase in behavioral performance. We speculate that the omission of reward-associated signals during a false alarm trial directs the animal's attention towards the newly rewarded texture. Elevated attentional signals have been shown to modulate sensory-driven responses in visual cortex[42]. Thus, the attentional signals may be read out by the value neurons, which in turn reshape the texture selectivity of surrounding neurons. Together, this may enhance sensory perception.

## Methods

**Animals**. C57BL/6J male mice (Janvier Labs) aged 6 weeks were group housed on a 12-h light cycle (lights on at 8:00 a.m.) with littermates until surgery. Two weeks after surgery, mice kept under standardized conditions at the animal facility of the university of Geneva, with an inverted light-dark cycle 7–8 days before the first training session. The behavioral experiments were performed during the dark phase. All procedures were conducted in accordance with the guidelines of the Federal Food Safety and Veterinary Office of Switzerland and in agreement with the veterinary office of the Canton of Geneva (licence numbers GE/28/14, GE/61/17, and GE/74/18). C57BL/6J male mice (Janvier Labs) aged 6 weeks were group

housed on a 12-h light cycle (lights on at 8:00 a.m.) with littermates until surgery. Two weeks after surgery, animals were housed under standard conditions, with an inverted light–dark cycle 7–8 days before the first training session.

**Surgery and intrinsic optical imaging**. Stereotaxic injections of adeno-associated viral (AAV) vectors were carried out on 6-week-old male C57BL/6 mice. A mix of $O_2$ and 4% isoflurane at 0.4 L min$^{-1}$ was used to induce anesthesia followed by an intraperitoneal injection of MMF solution, consisting of 0.2 mg kg$^{-1}$ medetomidine (Dormitor, Orion Pharma), 5 mg kg$^{-1}$ midazolam (Dormicum, Roche), and 0.05 mg kg$^{-1}$ fentanyl (Fentanyl, Sinetica) diluted in sterile 0.9% NaCl. AAV1-hSyn1-mRuby2-GSG-P2A-GCaMP6s (Penn Vector Core; 100 nl)[20] was delivered to L2/3 of the right barrel cortex in S1 at the approximate location of the C2 barrel-related column (1.4 mm posterior, 3.5 mm lateral from bregma, 300 μm below the pia). For long-term in vivo calcium imaging, a 3-mm diameter cranial window was implanted, as described previously[43].

Two weeks after surgery, the C2 barrel column was mapped again using intrinsic optical imaging to confirm the location of mRuby2/GCaMP6s expression. To do this, a mix of $O_2$ and 4% isoflurane at 0.4 L min$^{-1}$ was used to induce anesthesia followed by an intraperitoneal injection of MM solution consisting of 0.2 mg kg$^{-1}$ medetomidine and 5 mg kg$^{-1}$ midazolam diluted in sterile 0.9% NaCl. The C2 whisker was inserted into a capillary connected to a piezo actuator. Intrinsic signal was collected during repeated whisker stimulation (1 s at 8 Hz). A 100-W halogen light source connected to a light guide system with a 700-nm interference filter was used to illuminate the cortical surface through the cranial window. Reflectance images 300 μm below the surface were acquired using a ×2.7 objective and the Imager 3001F (Optical Imaging, Mountainside, NJ) equipped with a 256 × 256 pixels array charge-coupled device (CCD) camera (using VDaq software). The built-in Imager 3001F analysis program (Winmix software) was used to visualize the responses and produce an intrinsic signal image by dividing

the stimulus signal by the pre-stimulus baseline signal. An image of the vasculature was then acquired using a 546-nm interference filter, and superimposed on the intrinsic signal image. This reference image was used later to select an appropriate field of view (FOV) using 2PLSM. After this procedure, a metal post was implanted laterally to the window using dental acrylic to restrict head movement during behavior and imaging.

**Habituation and water deprivation**. Mice were handled and accustomed to be head restrained on the training setup for 10–15 min over 4–5 days. Water deprivation started 3–5 days before the first training session and discontinued at the end of the training. Weight was monitored daily during this period and the amount of water given was adjusted to prevent them from losing more than 15% of their original weight. Altogether, mice received a minimum of 1 ml of water per day corresponding to the amount they drank during the training as rewards plus the amount that the experimenter provided outside of the training sessions.

**Texture discrimination task**. Mice were trained to discriminate between two commercial-grade sandpapers (P120 and P280) in a Go/No-go paradigm as described previously[5]. The control of the devices and the recording of behavioral parameters were performed using a data acquisition interface (PCI 6503, National Instruments) and custom-written LabWindows/CVI software (National Instruments). Licks were detected electrically. Mice remained on a metallic plate that maintained an electrical potential difference with the licking spout. The electrical circuit was closed when mice touched the spout with their tongue, producing a 1.2-μA current that was detected by the acquisition interface. Whisking activity was measured with an optical barrier that detected the changes in intensity when whiskers swept through. To achieve this, an 850-nm LED beam was used as light source (HIR204C, Everlight Electronics) and an 860-nm phototransistor (PT 202C, Everlight Electronics) was used to detect intensity variations through a 1-mm hole placed 60 mm away from the light source, at a sampling frequency of 10 kHz. Whisking activity was quantified as the frequency at which individual whiskers crossed the light beam placed ~1 mm in front of and centered on the presented texture. The licking and whisking rates were calculated as the average number of events over a sliding window of 100 ms and normalized per second.

Sandpapers were attached to a four-arm wheel (2 × 2 of the same sandpapers) mounted on a stepper motor (T-NM17A04, Zaber) and a motorized linear stage (T-LSM100A, Zaber) to move textures in and out of reach of the whiskers. At the start of each trial, the wheel spun for a random amount of time while in the rear position of the linear stage (approximately between 0.5 and 1 s) and stopped between two textures positions. To present a texture the linear stage first moved to the front position and then the stepper motor rapidly slid the sandpaper into the whisker's reach at ~15 mm from the snout with an angle of 70° relative to the rostro-caudal axis. In the first phase of the training, the coarser P120 sandpaper was the rewarded texture (i.e. the target stimulus for which the mouse was incited to lick the spout in order to receive a water reward) and the P280 sandpaper was the non-rewarded texture (i.e. the non-target stimulus for which the mouse was incited to refrain from licking the spout). Initially, mice were trained to trigger a 4–6-μl sucrose water reward (100 mg/ml) by licking the spout during the presentation of the P120 texture (rewarded). Then, they were gradually familiarized with the P280 texture presentation (non-rewarded), from 0 to 30% of the trials, within two sessions (one session per day, 150–300 trials per session). Imaging started when P120 and P280 textures were pseudo-randomly presented with 50% probability for each trial type with a maximum of four consecutive presentations of the same stimuli. A trial consisted of a 1-s pre-stimulus period followed by a 3-kHz auditory cue for 200 ms, a delay period of 500 ms after which the texture reached the whiskers within 150 ms and remained there for 2 s before being retracted. Licking during the P120 texture presentation triggered a water reward at the end of the 2-s presentation, and the corresponding trial was scored as a 'hit'. Licking during the P280 texture presentation triggered a 500-ms white noise sound exposure at the end of the 2-s presentation plus a 5-s time-out period, and the trial was scored as a 'false alarm' (FA). In the absence of a lick during stimulus presentation, trials were scored as a 'miss' or a 'correct rejection' (CR) for P120 and P280 stimuli, respectively. To prevent the mice from compulsive licking during training, in addition to the aforementioned rules, mice had to show a 2-fold increase in the licking rate during stimulus presentation as compared with the pre-stimulus baseline period to get rewarded on the P120 texture presentation. Around 250–400 trials per session were performed (1 session per day) at a rate of ~6 trials/min.

The overall performance of the animal was calculated as the percentage of correct trials (hits + CRs) over an entire session or over a sliding window of 200 trials. The hit and FA rates were calculated as $N_{hit}/(N_{hit}+N_{miss})$ and $N_{FA}/(N_{FA}+N_{CR})$ respectively where $N$ is a number of trials for an entire session or over a sliding window of 200 trials. Mice were considered experts when the average performance per session reached a level of 70% correct trials (the expert criterion) over two consecutive sessions. In the second phase of training (i.e. reversal learning), reward contingencies were inverted (i.e. the P280 texture was rewarded whereas the P120 texture was not) and mice were trained until they reached the same expert criterion again in two consecutive sessions.

**2PLSM**. We used a custom built 2-photon laser scanning microscope mounted onto a modular in vivo multiphoton microscopy system (https://www.janelia.org/open-science/mimms-10-2016) equipped with an 8-kHz resonant scanner and a ×16 0.8NA objective (Nikon, CFI75), and controlled with Scanimage 2016b[44] (http://www.scanimage.org). Fluorophores were excited using a Ti: Sapphire laser (Chameleon Ultra, Coherent) tuned to $\lambda = 980$ nm that was slightly underfilling the back aperture of the objective to extend the depth of field to 5 μm. Fluorescent signals were collected with GaAsP photomultiplier tubes (10770PB-40, Hamamatsu) separating mRuby2 and GCaMP6s signals with a dichroic mirror (565dcxr, Chroma) and emission filters (ET620/60 m and ET525/50 m, respectively, Chroma). Fast volumetric imaging was performed at 11.5 Hz using a piezo z-scanner (P-725 PIFOC, Physik Instrumente) for moving the objective over the z-axis. Each acquisition volume consisted of 5 contiguous planes (with 5-μm steps between planes) of 400 × 400 μm (512 × 256 pixels) allowing post-hoc z-motion correction which may be generated by licking-induced brain motion artifacts[21].

**Image processing**. Images were processed using custom-written MATLAB scripts and ImageJ (http://rsbweb.nih.gov/ij/). Lateral and axial motion corrections were performed using the mRuby2 signal as a reference. First, rigid lateral movement vectors were calculated based on individual trial movies from the average z-projection of the 20-μm imaged volumes using the NoRMCorre MATLAB tool-box[45]. Residual bidirectional scanning artifact vectors were calculated using a highest-pixel-line signal correlation between the two scanning directions on the entire frame. Inter-trial registration was calculated using a custom-written cross-correlation algorithm based on the rigid image stack registration plugin in ImageJ. All calculated lateral motion corrections were applied on both the mRuby2 and GCaMP6s signals. Second, axial motion correction was performed using cross-correlation on linearly interpolated volumes (with a factor 3). The image planes with the highest correlation to a reference image, defined as the center image plane of the first volume, were selected. For an unbiased extraction of the GCaMP6s fluorescence signals from individual neurons, regions of interest (ROIs) were drawn manually for each session based on neuronal shape using the mRuby2 signal. The fluorescence time-course of each neuron ($F_{measured}$) was measured as the average of all pixel values within the ROI. Local neuropil signal ($F_{neuropil}$) was measured for each ROI as the average of pixel values within an automatically defined ring of 15 μm width, 2 μm away from the ROI (excluding overlap with surrounding ROIs)[46]. The fluorescence signal of a cell body was then estimated as $F(t) = F_{measured}(t) - r \times F_{neuropil}(t)$ with $r = 0.7$[47]. Residual trends were removed by subtracting the 8th percentile of each trial[48]. Normalized calcium traces $\Delta F/F_0$ were calculated as $(F-F_0)/F_0$, where $F_0$ is the median of the individual mean baseline fluorescence signal of each trial over a 1-s period before the start of the stimulation. For individual stimulation sessions (see Individual stimulation session and neuron categorization section) and spontaneous activity recordings, $F_0$ is the 30th percentile of each trial trace. For display, traces were additionally filtered with a Savitzky-Golay function (2nd order, 500-ms span).

**Activity onset analysis**. Normalized calcium traces ($\Delta F/F_0$) were aligned to either the onset of the texture presentation or to the first lick during the texture presentation for each neuron across all hit trials of an expert session. For both realignments, the onset of the neuronal response was calculated as the time, relative to the texture or first lick onset, at which the average of the response reached half of its maximum amplitude.

**Individual stimulation session and neuron categorization**. Prior to the start of the training, nine mice were imaged in the experimental training configuration, where task-related stimuli were presented independently of one another in a pseudo-random fashion. Data acquisition was organized in trials of 10 s, each starting with a 3-s baseline after which one of the following conditions was presented at a random time within a 4-s window: 2-s texture, 0.2-s sound (auditory cue) or water valve opening to incite licking, and finishing with another 3-s of recording. In 20% of the trials, no stimulation was applied. Whisking and licking events were recorded over the course of the session.

To determine if neuronal activity was significantly modulated by texture or sound stimuli, we compared, for each neuron across trials, the average normalized fluorescence over 1 s before and after the stimulus onset using a paired-sample $t$-test at a significance threshold of 5%. To account for noise in our data due to possible stimulation-induced movement artefacts, we performed the same test using the mRuby2 signal. None of the neurons showed a significant change in mRuby2 signal upon texture and sound stimulation.

We used a random forests machine-learning algorithm to decode behavioral features (licking and whisking rates) from the activity of single neurons. This procedure allowed us to categorize single neurons as either decoding whisking, licking, or both. Given the slow kinetics of calcium transients captured by the GCaMP6s sensor, spiking rates were inferred from the $\Delta F/F_0$ trace and used as input to the algorithm, which allowed to temporally match behavioral event variations (i.e. whisking or licking rates) to neuronal activity. Firing rates at each imaging frame were inferred from normalized calcium traces ($\Delta F/F_0$) using a fast nonnegative deconvolution method (https://github.com/jovo/oopsi)[49] with variable

background fluorescence estimation and a $K_d$ of 144 nM[50]. In order for the algorithm to capture differences in activity levels between neurons, all trial traces of all neurons recorded per mouse were concatenated before inferring spikes. To account for putatively preceding pre-motor and/or following sensory-related activity in S1 relative to behavioral events, the neuronal activity traces were shifted negatively and positively in time with a maximum shift of 500 ms. Eleven time bins of inferred firing rates (discretized in time bins of 100 ms) centered on zero time-shift were used to predict instantaneous behavioral features and composed a vector $X_i(t) = [x_i(t - 500\,\text{ms}), \dots x_i(t), \dots , x_i(t + 500\,\text{ms})]$ where $x_i(t)$ represents the inferred firing rates of the $i$th neuron at zero time-shift. Licking and whisking rates were down sampled to 11.5 Hz in order to temporally match calcium imaging data. The ranger function of the ranger R package version 0.10.1 was used to construct regression forests, with each behavioral feature as dependent variable and the binned inferred firing rates of a given neuron as predictors. For each neuron, two regression forests were constructed, one to decode whisking and the other licking. Most arguments of the function were kept at default settings, except the following: the number of trees was set to 128, the minimum size of terminal nodes was set to 2, the number of predictor variables randomly sampled at each node split was set to the maximum between 1 or the third of the number of predictors, and the variable importance mode was set to "impurity". To obtain a prediction for all trials, 5-fold cross-validation was applied by training the algorithm on 80% of the trials (i.e. training set) and evaluating it on the remaining 20% of the trials (i.e. test set). Since data acquisition was discretized by trial, for each cross-validation the training and test set trials were concatenated for training and prediction, respectively. For each neuron and for each behavioral feature, the decoding accuracy was assessed by computing the Pearson's product-moment correlation coefficient between the observed and predicted behavioral event levels. In order to get an estimate of the noise in the prediction levels, the same analysis was performed using the mRuby2 signal as a control. Neurons were classified as decoding a given behavioral feature if their Pearson's correlation coefficient computed on the GCaMP6s signal was five standard deviations away from the mean of the Pearson's correlation coefficients for all neurons computed on the mRuby2 signal. Neurons meeting these criteria for both whisking and licking were classified as decoding both behavioral features.

**Spontaneous activity correlation**. Spontaneous calcium transients were recorded for 10 min after mice reached the expert level before and after texture reversal. Pairwise Pearson's correlation coefficients were calculated on the normalized calcium traces.

**Discrimination and choice indices**. The selectivity of each neuron was expressed by a Discrimination index (DI) that was calculated based on neurometric functions using a receiver-operating characteristic (ROC) analysis[22,51,52]. Normalized mean calcium signals ($\Delta F/F_0$) during the 2-s stimulus presentations in the P120 texture trials were compared to the P280 texture trials. ROC curves were generated by plotting, for all threshold levels, the fraction of P120 trials against the fraction of P280 trials for which the response exceeded threshold. Threshold levels were defined as a linear function from the minimal to the maximal calcium signals. DI was computed from the area under the ROC curve (AUC) as follows: DI = (AUC −0.5) × 2. DI values vary between −1 and 1. Positive or negative values indicate larger or smaller responses to P120 than to P280 texture presentations, respectively. Statistical significance of the measured DI value was assessed by performing a permutation test, from which a sampling distribution was obtained by shuffling the texture labels of the trials 10,000 times. The measured DI was considered significant when it was outside of the 2.5th–97.5th percentiles interval of the sampling distribution. For the choice index (CI), the same calculation was performed, with the difference that trials in which the animal licked during the texture presentation were compared to trials with no lick. For building the temporal evolution of the DI and CI across reversal learning, both indices were calculated over a sliding window of 100 trials every 5 trials.

**Calcium signals relative to behavioral strategies**. For all hit trials of an expert session, average whisking and licking rates were calculated as the average number of events over the entire texture presentation window. For each mouse, the median value in both distributions was used to separate low and high whisking or licking rate trials.

**Error history**. Error history for each neuron was calculated as the normalized difference between the average calcium signal during hit trials ($\bar{R}$) over a sliding window of 200 trials as follows:

$$\text{Error history}(t) = \frac{\bar{R}_{\text{hit(post FA)}}(t - 100 : t + 100) - \bar{R}_{\text{hit(post hit)}}(t - 100 : t + 100)}{\bar{R}_{\text{hit}}(t - 100 : t + 100)}$$

where $R_{\text{hit(post FA)}}$ is the calcium signal in a hit trial that was preceded by a FA trial, and $R_{\text{hit(post hit)}}$ is a calcium signal in a hit trial that was preceded by another hit trial. $R_{\text{hit}}$ is the calcium signal in any hit trial, and $t$ is the trial number, relative to the trial at which behavioral performance reaches the expert criterion. To estimate the fraction of neurons with an error history above chance, all hit trials within each window of 200 trials were randomly permuted for each neuron, replacing $R_{\text{hit(post FA)}}$, $R_{\text{hit(post hit)}}$, and $R_{\text{hit}}$ in their respective trial positions. Then, an error history value was calculated based on the permuted data set. This process was repeated 1000 times to obtain 95% confidence intervals for each observed error history value.

**Immunohistochemistry**. Post-hoc immunohistochemistry of GABA was performed on mRuby2/GCaMP6s-expressing neurons. In all, 100-μm-thick tangential sections were produced using a vibratome (Leica VT 1000). The sections were washed 3 × 3 min in 500 μl Tris-buffered saline (0.1 M Tris, 150 mM NaCl) containing 0.1% Tween (TBST), then pre-treated with TBST and 0.1% Triton-X for 20 min followed by a 3 × 3 min TBST wash. They were blocked in 300 μl TBST containing 10% normal donkey serum (ab7475, Abcam) for 1 h and incubated with mouse anti-GABA antibody (ab86186, Abcam) diluted 1:500 for 72 h at 4 °C. After another 5 × 3 min wash in TBST they were incubated in 300 μl of donkey anti-mouse antibody coupled to Alexa Fluor 647 (A32787, Thermo Fisher Scientific) diluted 1:200 in TBST for 1 h at room temperature. Finally, they were washed 10 × 3 min in TBST and then for 1 h in PBS before being mounted onto glass slides. We applied Fluoroshield mounting medium with DAPI (Abcam) before applying the coverslip. The sections were imaged using a Zeiss Confocal LSM800 Airyscan.

**S1 inactivation and whisker trimming**. To inactivate S1, the GABA (G-amino-butyric acid) agonist muscimol was injected in a separate set of expert mice ($N = 5$ mice; this data set was also used as a control group for another study[53]). During the test session, high baseline performance (>70%) was first recorded for 100 trials before the injection was performed. Under light anesthesia (4% isoflurane at $0.4 \text{ L min}^{-1}$), a small hole was drilled through the imaging window above the previously mapped C2 barrel column to provide access to a glass pipette through which 300 nl of Muscimol (Bodipy-TMR-X, 5 mM in cortex buffer with 5% DMSO, Thermo Fisher Scientific) was injected at 300 and 500 μm below the pia. Mice were left to recover for 45 min and their behavioral performance was then assessed for another 100 trials. For the whisker trimming experiment, a similar baseline performance was first recorded for 100 trials before trimming the whiskers on the side of the snout contralateral to the texture presentations, and tested the performance for 50 trials. This ensured that trimming itself did not alter performance. Then, the whiskers that were in contact with the textures (ipsilateral to the texture presentation side) were trimmed, and the effect on task performance was measured for another 50 trials.

**Statistics and reproducibility**. All statistics were performed using MATLAB. For all figures, significance levels were denoted as *$P < 0.05$, **$P < 0.01$, ***$P < 0.001$, and ****$P < 0.0001$. No statistical methods were used to estimate sample sizes. All comparison tests were performed two-sided. Non-parametric tests were used for sample sizes smaller than 15. For the training experiments, the fields of view across mice were of similar quality and the number of neurons recorded ranged between 42 and 113. For immunostainings, 2–3 fields of view per mice of similar quality containing 110–201 neurons were analyzed.

**Reporting summary**. Further information on research design is available in the Nature Research Reporting Summary linked to this article.

## Data availability

The data used to generate the figures is freely available at the CERN data repository Zenodo https://zenodo.org/communities/holtmaat-lab-data/ with https://doi.org/10.5281/zenodo.3824493.

## Code availability

The principal Matlab code that was used for data analysis is freely available at the CERN data repository Zenodo https://zenodo.org/communities/holtmaat-lab-data/ with https://doi.org/10.5281/zenodo.3824493.

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

## Acknowledgements

We thank Ariel Gilad for advice on the behavioral paradigm; Fritjof Helmchen and Abhishek Banerjee for discussions and suggesting the error history analysis; Jose Manuel Nunes for advice on assessment of selection criteria in the prediction model; Sebastien Pellat for technical support and engineering; Laura Bussien and Elodie Husi for assistance with histology; Tobias Rose for making available the mRuby-GCaMP constructs; Sonja Hofer for sharing viral vectors; and Pieter Roelfsema for useful comments on the manuscript. This project was supported by the Swiss National Science Foundation (grants 31003A-153448, 31003A_173125, CRSII3_154453, and NCCR Synapsy 51NF40-158776), and a gift from a private foundation with public interest through the International Foundation for Research in Paraplegia.

## Author contributions

R.C., S.P., and A.H. designed the experiments. R.C. and T.B. performed the experiments. S.P. and R.C. designed and built experimental setups and analysis software. R.C. analyzed the data and L.F. performed random forests modeling. A.C. and A.H. provided equipment and technical expertize. A.H. supervised the research. R.C. and A.H. wrote the manuscript. All of the authors edited the manuscript.

## Competing interests

The authors declare no competing interests.
