## [Peer Review File · Nature Communications]

Reviewers' Comments:

Reviewer #1:

Remarks to the Author:

Chereaux et al investigate the neural representations of sensory input and choice behavior using Ca imaging in mouse S1 during the learning of a texture discrimination task and the effect of reversal of reward contingencies on these neural representations.

The most important new contribution of this study is in the longitudinal investigation of the trajectory of neural responses in primary sensory cortex during reversal learning. The experiments are generally well-designed and the data seem of high quality. The most interesting finding is that there are value-sensitive neurons that can be identified in primary sensory cortex (as opposed to value/reward-reporting neurons only in higher-order areas), and that these neurons show a transient increase in activity at the onset of learning *prior* to circuit remapping. This is an exciting contribution to our understanding of what is encoded in primary sensory cortex, and how it can dynamically change under altered behavioral contingencies. These data also show that reward-based training increases the number of P120 responsive neurons (Figure 1j) without necessarily reducing the number of P280 selective neurons, and that these response categories apparently can be flexible with reversal learning. Overall the manuscript would be much improved with a more focused analysis of specific response categories for the initial training and then during/after reversal learning that are separated in different figures; the figures were exceedingly difficult to parse. In addition, color is used too enthusiastically – the manuscript is very hard to read if you are color-blind and there are more than a dozen different colors and categories presented, and different figures use entirely different color schemes.

1. Figure 3 is an important control, showing that texture representations in S1 do not have some inherent preference for lower frequencies. The better control here is showing that the initial training with P280 reward leads to the same remapping as when the initial training is to P120 reward. Was this ever done? Were the results different? I am not convinced that reversal learning will help determine whether there is an inherent preference for coarser textures. The authors should revise this justification, unless they carry out the above experiment (start training with the fine texture as the rewarded one.)

2. The reversal training is a very interesting paradigm that addresses important questions about remapping during learning. However, this figure is exceedingly difficult to read. First, the schematic of reward contingencies from Fig. 1 should be repeated here, to keep things parallel. Second, there are too many colors in the figure, and it is not necessary for the data presentation (in fact, it is confusing). The heat maps for Fig 3c could be presented in black and white without any loss. It would be helpful to keep the organization and format the same as Fig.1 throughout – i.e. the number of examples in Fig 1f should match. Figure 3d is excessively complex, and I could not make heads nor tails of it. There are too many colors. The data are digested substantially before they are plotted, and there are too many categories shown at once. Red and green histograms with an intermingled brown is difficult for people to see, even if they are not colorblind. Can the figure be separated into 2 different ones?

3. The shift in texture selective neurons during reversal learning is interesting and merits its own figure. This would help the reader considerably in evaluating the relative remapping that has occurred, since texture is the primary variable that is changed during reversal learning. Suggestion: please present evolution/shifts in texture preference in one figure and value/choice into another? This would help with readability. I am particularly interested in the neurons that change their texture preference over training and retraining – was there any thing about the activity of those neurons that would help predict their lability?

4. Figure 4 is a critical figure for the manuscript but was difficult to interpret. Specific suggestions are: 1 4f – please make it clear that these are three different animals. Why were these different neural classes selected from 4 different animals? The information conveyed by the different learning

trajectories is really interesting, because it shows that texture/value selectivity changes can precede behavioral changes.

2 In figure 5 – Fig 5e,f is very interesting, but the data that are plotted are difficult to understand.

What are the units of error history? How can it be a continuous variable? Isn't a trial either FA prior to Hit, or a FA prior to FA, or Hit prior to Hit? Is this an average across the entire population of neurons? What was the n/neurons for each animal? Was this only for the 10 animals that showed value selective neurons? Please provide a supplemental table that shows the cell n's for the number of value selective neurons per animal (non-selective and choice selective can also be included). If this changes across trials, it could be binned for each 50 trials.

5. The paper would be substantially improved by a pie chart that indicates the average distribution of neuron types (P120, P280, value, choice, and non-selective) in naïve versus expert vs early reversal vs expert reversal animals. This is also presented in Fig 2i, but there are so many different categories shown in that paper that it is hard to interpret and compare the data.

6. Although it was admirable that the authors look at the fraction of neurons that were GABAergic, the analysis was ultimately incomplete. The authors report that only about 6% of labeled neurons were GABAergic, but even a small number of interneurons could be important if overrepresented in one of their activity-defined groups (texture, value, choice, etc.). It is also unclear whether the histologic images were taken in the field of view used for calcium imaging, or some other area.

7. A summary figure/model would help considerably in conveying the main message of the paper.

Minor.

1. It is very interesting that only 10/12 animals showed value selective neurons. What might be possible reasons for this?

2. Figure 1i – please indicate what grey shaded data indicate. What do red and green refer to? Please edit for clarity.

3. There are a lot of different colors used; it is hard to see the continuity between figures (red/green, orange/brown). Please select one color scheme and stick with it throughout the text. For example, Fig 2g,h has red, green, and brown, but brown here indicates something different than in other figures.

4. Figure 4e – this is potentially very helpful, but it is unclear why blue is 0. Could the colors used for different classes be synchronized across figures? Is this just after reversal learning?

5. In Sup Fig 1, why does licking rate max out at 1Hz? Animals can lick at much higher rates. Please explain. Is it lick probability, not Hz?

6. In paragraph 1 of Results, the authors justify S1 silencing with muscimol by saying "to confirm that S1 is required for this task..." Based on Hong and Bruno's experiments where transient silencing of S1 interrupted detection task performance, but lesioning S1 caused compensation elsewhere in the circuit and thus did not, it seems plausible to me that transient silencing with muscimol could interrupt the discrimination task because S1 is used in the task, but maybe not required under all circumstances. Therefore might it be safer to say that S1 is used? This may be a minor semantic issue.

7. In the Reporting Summary, the authors do use a Pearson correlation to assess whether the neurons encode whisking and licking, but do not discuss effect sizes, so this check box should not be N/A.

8. In the legend for Supplemental Fig. 2a (example of mRuby2 expression, DAPI staining, and GABA immuno) the text reads "Note that the vast majority of GABA-positive neurons is negative for GABA and vice-versa." This should read "the vast majority of mRuby2-positive neurons"?

Reviewer #2:

Remarks to the Author:

Comments to the Authors:

Chéreau, Bawa, Fodouliau, Carleton, Pagès and Holtmaat examines the effects of reward contingency reversal in mouse primary somatosensory cortex neurons using a Go/No-Go texture discrimination task. Mice were trained to the 'go/no-go' texture discrimination task and neuronal responses were monitored using GCaMP6s and 2PLSM. Upon finding a bias for P120 texture-selective neurons when pairing the P120 texture with reward, the group addressed whether this bias was due to an intrinsic bias toward coarser textures, or whether the bias is due to population encoding of higher order features, by switching the rewarded texture from P120 to P280. Ultimately, it was found that some neurons switched their selectivity to the P280 texture, implying that these neurons remained selective to the rewarded 'go' texture. While many neurons were considered to gain or lose a heightened discrimination response between textures, the authors did find some cells whose activity appears to encode the same information (value, identity, or choice) across contexts. Further analysis revealed that the response of value-selective neurons changes prior to performance improvements based on error history.

It builds upon recent studies examining texture discrimination during learning, by asking whether those responses were stable across different contexts. The experiments and analysis are both solid and thorough. Overall, this investigation is very useful in that it elucidates possible features encoded by S1 that were initially thought to be exclusively 'higher order', including stimulus associated reward value or behavioral choice and it further suggests that an a priori preference for a specific texture in S1 may not exist.

Major Points

While there are several interesting findings reported in the manuscript, the main issue is clarity. The way it is presented is very confusing. I can understand the author's logic in starting with broad categorical breakdowns of functional types and sub-dividing these classes as the text proceeds, but I think this logic is distracting to the main points of the manuscript. Further, references to "selectivity", "go/no go", "behavior", "choice", "reward" are expressed in a way that is also confusing because their definition shifts through the course of the manuscript.

1) The authors spend Figs. 1-3, examining P120-vs.P280-selectivity and Go/No-Go selectivity from the perspective of pre-reversal and then post-reversal learning and then re-analyzes these responses with respect to texture-selective, choice-selective, and value-selective neurons in S1 (Fig 4-5.). This is disorienting for the following reasons:

a) P120-vs.P280-selectivity and Go/No-Go selectivity are dynamic definition which changes depending on pre- and post-reversal learning conditions and thus makes it difficult for the reader to keep track of what the author is referring to.

b) P120-vs.P280-selectivity and Go/No-Go selectivity actually reflect some combination of texture, choice, and value selective neurons. After providing a sense for their distribution and breakdown in Figs. 2i, and 3d-h, it is not clear how these same neurons map onto the texture-selective, choice-selective, and value-selective neurons reported in the later figures?

b) It makes more sense to define and report on texture-selective, choice-selective, and value-selective neurons (the main points) from the beginning of the manuscript. These are "static" definitions that don't change depending on the pre-reversal or post-reversal. The fraction of neurons for each of these functional cell classes and how they change over the course of reversal learning should then be presented. Shifting to P120-vs.P280-selectivity and Go/No-Go selectivity should be avoided as much as possible.

2) Similarly, the authors need to be more careful in how they discuss licking, whisking, and choice as it relates to behavior. The word behavior is used multiple times and very loosely applied to different things in a manner that also makes things confusing. Terms should be more clearly and conservatively defined. For example, there is a section on "L2/3 neurons selectivity report sensory input and not

behavioral output". Behavioral output constitutes many things. Only observed motor parameters that include licking and whisking have been analyzed while other motor parameters have not. The authors suggest that "licking behavior" is not encoded in S1 but "choice behavior" is. Yet, licking in this task is the motor behavior that expresses the animal's choice.

In fact, what the authors are describing are paired stimulus associations. From this standpoint "texture-selective" neurons can be referred to as "stimulus-only", "choice selective" can be referred to as "stimulus-response" neurons and "value selective" can be referred to as "stimulus-outcome" neurons.

Minor Points:

- There are a few sweeping statements that don't seem to have a clear basis for support. For example, "even though a preference is probably absent at the population level for the textures that we used"
- A note on how long it took to reach expert criterion after reversal would have been appreciated
- The caption of Supplementary Figure 2 ("Note that the vast majority of GABA-positive neurons...") is not worded clearly.

Response to Reviewers' comments: (response to reviewers in blue)

General response:

We thank the reviewers for their comments and suggestions. These have been very valuable to us for increasing the interpretability of the data and readability of the manuscript. While evaluating the reviewers' comments we realized that we had been inconsistent in the presentation of the neuronal selectivity data, i.e. at some times we presented them as go/no-go cue selective and at other times as P120/P280 selective. Obviously, as the reviewers noted, this became highly confusing at the point where reversal learning was discussed. Therefore, in the revised version, we opted to consistently present the selectivity of neurons according to the texture that was presented (i.e. P120/P280). We believe that this has greatly improved the narrative and takes away some of the confusion the reviewers did express.

Reviewer #1 (Remarks to the Author): Chereaux et al investigate the neural representations of sensory input and choice behavior using Ca imaging in mouse S1 during the learning of a texture discrimination task and the effect of reversal of reward contingencies on these neural representations. The most important new contribution of this study is in the longitudinal investigation of the trajectory of neural responses in primary sensory cortex during reversal learning. The experiments are generally well-designed and the data seem of high quality. The most interesting finding is that there are value-sensitive neurons that can be identified in primary sensory cortex (as opposed to value/reward-reporting neurons only in higher-order areas), and that these neurons show a transient increase in activity at the onset of learning *prior* to circuit remapping. This is an exciting contribution to our understanding of what is encoded in primary sensory cortex, and how it can dynamically change under altered behavioral contingencies. These data also show that reward-based training increases the number of P120 responsive neurons (Figure 1j) without necessarily reducing the number of P280 selective neurons, and that these response categories apparently can be flexible with reversal learning. Overall the manuscript would be much improved with a more focused analysis of specific response categories for the initial training and then during/after reversal learning that are separated in different figures; the figures were exceedingly difficult to parse. In addition, color is used too enthusiastically – the manuscript is very hard to read if you are color-blind and there are more than a dozen different colors and categories presented, and different figures use entirely different color schemes.

We thank the reviewer for her/his enthusiasm and for valuing the work as “an exciting contribution to our understanding of what is encoded in primary sensory cortex”. We have addressed her/his comments and suggestions, which we believe have substantially improved our manuscript.

1. Figure 3 is an important control, showing that texture representations in S1 do not have some inherent preference for lower frequencies. The better control here is showing that the initial training with P280 reward leads to the same remapping as when the initial training is to P120 reward. Was this ever done? Were the results different? I am not convinced that reversal learning will help determine whether there is an inherent preference for coarser textures. The authors should revise this justification, unless they carry out the above experiment (start training with the fine texture as the rewarded one.)

Fig. 3e (New Fig. 4d) shows that reversal training prompts a switch in the texture selectivity bias from P120 to P280. This suggests that at the population level, neurons do not have a major intrinsic preference for

one or the other texture. That said, we agree with the reviewer that this is best tested in naïve mice by training them on P280 texture-cued reward instead of a P120 texture-cued reward. In order to address this issue, we went back to our dataset that we obtained for **Fig. 2d-h**, and for which we had exposed mice to the various task-related stimuli prior to training. In this session, all the stimuli were presented separately, without a structure, and without a reward (apart from occasional rewards that were not cued – see **Fig. 2d**). From this dataset we extracted for each neuron its response to the P120 and P280 texture, and calculated the respective DI (**Reviewer Fig. 1**). This shows that there was a non-significant difference between the percentage of P120 or P280-selective neurons. This difference remains non-significant during the naïve phase of learning (as was shown in **Fig. 1j**, and **Reviewer Fig. 1**), and only shapes up to become significant during the expert phase. We believe that this provides sufficient evidence that the neuronal population that we tracked over learning was not a priori preferring one or the other texture. We have added this panel to Fig. 2 (**New Fig. 2f**).

Reviewer Figure 1: Left, fractions of P120 (brown) and P280 (orange) selective neurons measured from the touch neurons in the random stimulus session prior the initial training (N=8 mice, 549 neurons, P120 selective neurons: $1.5 \pm 0.7\%$; P280 selective neurons: $3.5 \pm 1.2\%$ Wilcoxon rank sum test, $P=0.27$). Right, for comparison, selective fractions in the naïve and expert phases of the initial training (Data shown in Fig. 1j; N=12 mice, 875 neurons, Wilcoxon rank sum tests, P120 selective neurons: naïve, $9.5 \pm 2.6\%$, expert, $19.2 \pm 4.2\%$, $*P=0.02$; P280 selective neurons: naïve, $5.3 \pm 1.1\%$, expert, $7.2 \pm 1.2\%$, $P=0.33$; P120 vs. P280 selective fractions: naïve, $P=0.36$, expert, $**P=0.006$). Bars and error bars represent averages and s.e.m.

2. The reversal training is a very interesting paradigm that addresses important questions about remapping during learning. However, this figure is exceedingly difficult to read. First, the schematic of reward contingencies from Fig. 1 should be repeated here, to keep things parallel. Second, there are too many colors in the figure, and it is not necessary for the data presentation (in fact, it is confusing). The heat maps for Fig 3c could be presented in black and white without any loss.

It would be helpful to keep the organization and format the same as Fig.1 throughout – i.e. the number of examples in Fig 1f should match.

Figure 3d is excessively complex, and I could not make heads nor tails of it. There are too many colors. The data are digested substantially before they are plotted, and there are too many categories shown at once. Red and green histograms with an intermingled brown is difficult for people to see, even if they are not colorblind. Can the figure be separated into 2 different ones?

We have completely revised this figure, and incorporated all of the reviewer’s suggestions.

The changes are the following:

A. We have separated **Former Fig. 3** into two figures (**New Fig. 3a-d**; and **New Fig. 4a-g**), and performed several additional changes in order to improve the readability (see point B-D).

B. New Fig. 3 now has the same organization as **Fig. 1**. We have repeated the schematic of the reward contingencies, and graphically displayed the reversal training paradigm in order to make the subsequent figures more intuitive. We provide 2 example neurons in **New Fig. 3d**, one that remains selective for the P120 texture [neuron 1] and one that reverses its selectivity to the P280 texture [neuron 2]. Neuron 2 represents the class that is further analyzed in **New Fig. 5**. Therefore, we felt that this is the most interesting type of selectivity to contrast with a neuron that does not change its selectivity. The number of example neurons in **Fig. 1f** has been reduced to simplify the figure.

C. We have improved the color code throughout **Fig. 3** as well as in the other figures, first, by changing the colors to black and white when colors were unnecessary for comprehension and/or were confusing. Second, the heat maps for **New Fig. 3d (Former Fig. 3c)** were changed from parula to magma which is defined as one of the “best at perceptual uniformity and is also robust to colorblindness” (cited from: <https://elifesciences.org/labs/c2292989/jetfighter-towards-figure-accuracy-and-accessibility>). We did try to turn the color maps into black and white, as per the reviewer’s suggestions, but this complicated distinguishing the small calcium variations from the background. Therefore, we opted for the ‘magma’ type of display.

D. The scatter plot that was displayed in **Former Fig. 3c** has now been simplified. Instead of combining post-reversal selectivity with the classification based on the quadrant position, we now first display neurons that are selective pre- and post-reversal, and simply label the quadrants according to the direction of the neuron’s preference (**New Fig. 4a**). We then further classify the neurons based on their selectivity before and after reversal learning, and divide them over 4 separate scatter plots (**New Fig. 4b**). We have also changed the naming of those classes of neurons, which we think should now be more intuitive to the reader (i.e. neurons that gained selectivity, lost selectivity, remained selective to the same texture or reversed their selectivity).

E. We now display the histograms that were attached to the scatter plot in **Former Fig. 3d** in a separate panel (**New Fig. 4c**).

3. The shift in texture selective neurons during reversal learning is interesting and merits its own figure. This would help the reader considerably in evaluating the relative remapping that has occurred, since texture is the primary variable that is changed during reversal learning. Suggestion: please present evolution/shifts in texture preference in one figure and value/choice into another? This would help with readability. I am particularly interested in the neurons that change their texture preference over training and retraining – was there any thing about the activity of those neurons that would help predict their lability?

We agree with the reviewer that the shift in selectivity is intriguing. As stated in our **general comment** to the reviewers, we realized that we had not consistently presented the data according to texture selectivity, which complicated interpretability. We have now improved the readability of the figures and main text by improving the description of the relative remapping that occurred during the reversal learning. By doing so, we believe that it is not necessary anymore to split the figure. That said, we did decide to present the evolution of the selectivity of value/choice neurons in a separate figure (**New Fig. 6**). This shows, indeed, that there is a period in which choice and value neurons display different responses from one another. This is then further characterized in **New Fig. 7**, by showing that value neurons display an error-history-dependent increase in activity during a discrete period of learning.

4. Figure 4 is a critical figure for the manuscript but was difficult to interpret. Specific suggestions are:
1 4f – please make it clear that these are three different animals. Why were these different neural classes selected from 4 different animals? The information conveyed by the different learning trajectories is really interesting, because it shows that texture/value selectivity changes can precede behavioral changes.

2 In figure 5 – Fig 5e,f is very interesting, but the data that are plotted are difficult to understand. What are the units of error history? How can it be a continuous variable? Isn't a trial either FA prior to Hit, or a FA prior to FA, or Hit prior to Hit? Is this an average across the entire population of neurons? What was the n/neurons for each animal? Was this only for the 10 animals that showed value selective neurons? Please provide a supplemental table that shows the cell n's for the number of value selective neurons per animal (non-selective and choice selective can also be included). If this changes across trials, it could be binned for each 50 trials.

We have made figure and text adjustments, following the reviewer's suggestions:

A. We have simplified the plot with examples of the evolution of the discrimination and choice indices (**New Fig. 6**). We have separated this figure from **Former Fig. 4** in order to introduce the more interesting analysis that is presented in **New Fig. 7**. To do this we focus on two interesting classes, the choice and value neurons, and highlight the transition period at which their discrimination and choice indices are shifting signs (or not). The two examples are now from the same animal.

B. The plots are presented as a continuous variable, since we calculated the average neuronal response over a sliding window of 200 trials for each type of sequence (**New Figure 7d**). To arrive at the Error History we used the following calculation: $\text{Error History} = [(\text{Average of all HIT trial Responses that were preceded by a FA trial}) - (\text{Average of all HIT trial Responses that were preceded by a HIT trial})] / (\text{Average of all HIT trial Responses that were preceded by a HIT trial})$ (**New Figure 7e**). Since the Error History represents a normalized difference between the 2 types of Hit responses it is unitless. For the data in **New Fig. 7e** all neurons of the same class from all mice were pooled and the curves represent the average of the error history functions of these neurons, aligned to the moment at which the mouse's performance reached the expert criterion. We performed the analysis on Hit trial responses since the theoretical framework suggests that FA-Hit sequences are instructive for learning, using the Hit-Hit sequences as a control.

C. We followed the reviewer's suggestion and now provide a table with the number of neurons in each class (i.e., gained, lost, texture, choice value and non-selective neurons) for all mice used in our dataset (**New Supplementary Table 1**). We refer to this table in the legend of **New Figure 7**.

5. The paper would be substantially improved by a pie chart that indicates the average distribution of neuron types (P120, P280, value, choice, and non-selective) in naïve versus expert vs early reversal vs expert reversal animals. This is also presented in Fig 2i, but there are so many different categories shown in that paper that it is hard to interpret and compare the data.

We apologize for the lack of clarity in our initial description of the analysis. All the information was present in the methods but the reviewer made us realize that this was requiring more explanations in the main text. The classification of the neurons was based on their selectivity for textures before and after reversal learning. For example, a neuron that was selective for the P280 texture before and after reversal training was considered to be a texture neuron (or a neuron that remained selective for the same texture). Similarly,

a neuron that was not selective prior to reversal, but selective for either the P120 or P280 after, was considered as a neuron that gained selectivity (conversely, neurons that lost selectivity were selective before reversal but not after). Choice or value neurons did reverse their selectivity to the other texture. Thus, a neuron's classification as being a 'texture', 'lost', 'gained', or 'choice/value' neuron, was inherited only after the reversal training. This is now summarized in **New Fig. 6a**.

6. Although it was admirable that the authors look at the fraction of neurons that were GABAergic, the analysis was ultimately incomplete. The authors report that only about 6% of labeled neurons were GABAergic, but even a small number of interneurons could be important if overrepresented in one of their activity-defined groups (texture, value, choice, etc.). It is also unclear whether the histologic images were taken in the field of view used for calcium imaging, or some other area.

We do agree with the reviewer. From our experiments, we cannot rule out that GABAergic neurons are represented in one of our neuronal classes or categories. Following the reviewer's suggestion, we now mention this possibility in the discussion. We also specify in **Supplementary Fig. 2** that the quantification was performed on brain sections from a subset of mice, which not necessarily represents the fields of view from the imaging experiments.

7. A summary figure/model would help considerably in conveying the main message of the paper.

We have added a schematic to **New Fig. 6a**, in which we lay out the various classes that can be defined after reversal learning. It indicates the various transitions in selectivity that we detected, together with their fractional presence in the population.

Minor.

1. It is very interesting that only 10/12 animals showed value selective neurons. What might be possible reasons for this?

Indeed, throughout our dataset, we did not find all classes of neurons represented across all mice. We cannot rule out that this was indicative of variability in perceptual feature representation across mice. We rather think that this was due to a limited neuronal sample size in each mouse. In choosing the field of view and pixel size, we had to compromise between having a good imaging resolution - to be able to reliably identify neurons across sessions - and capturing as many neurons as possible.

2. Figure 1i – please indicate what grey shaded data indicate. What do red and green refer to? Please edit for clarity.

The grey shaded area in **Fig. 1i** indicates the number of non-selective neurons. This is now stated in the figure panel. The green and red colors in **Fig. 1** used to refer to Go and No-go trials. In the new version of the figures, we have changed this, and now refer only to P120 and P280 texture selectivity.

3. There are a lot of different colors used; it is hard to see the continuity between figures (red/green, orange/brown). Please select one color scheme and stick with it throughout the text. For example, Fig 2g,h has red, green, and brown, but brown here indicates something different than in other figures.

We apologize for the confusion that our color scheme may have caused. We have greatly simplified it by comparing the responses and selectivity only between the 2 textures instead of their reward contingencies (Go and No-go cues). The color scheme is maintained throughout the paper, i.e. dark

brown for P120 selectivity and light brown for P280 selectivity. Blue for neurons that gained selectivity after reversal, and green for neurons that lost selectivity. Red indicates that selectivity was reversed. Within the latter population, purple or orange refers to the 'choice' and 'value' neurons, respectively.

4. Figure 4e – this is potentially very helpful, but it is unclear why blue is 0. Could the colors used for different classes be synchronized across figures? Is this just after reversal learning?

We chose to indicate a fraction of 0 in blue (in **Former Fig. 4e**, **New Fig. 5e**) in order to make this value very distinct from those fractions that were close to 0. This visually strengthens the point that the selectivity shift in 'value' or 'choice' neurons does not originate from a change in licking behavior. Only a negligible fraction of the neurons throughout all classes (value, choice, texture, gained and lost) fell into the category of 'licking neurons' (from **Fig. 2**). The classes represent the post-reversal expert phase, whereas the categories were determined in a pre-training session in which cues were randomly presented, and whisking and licking behavior was quantified (see **Fig. 2**). We have also improved the consistency of the colors for the different classes across figures.

5. In Sup Fig 1, why does licking rate max out at 1Hz? Animals can lick at much higher rates. Please explain. Is it lick probability, not Hz?

Indeed, mice often lick at much higher frequencies to trigger the reward. However, we felt that displaying licking rates as a continuum up until the highest values was not very informative, since these high licking rates do, in our opinion, not represent a relevant parameter in the task. Instead, we wanted to emphasize the difference between trials where the mouse licked or did not lick (i.e. which better reflects the performance of the mouse) as well as the onset of licking relative to when the texture is presented. Therefore, in **Supplementary Fig. 1**, we chose to display licking rates higher than 1 Hz in plain white, which nearly binarizes the lick behavior and draws the viewers' attention more to the lick onset.

6. In paragraph 1 of Results, the authors justify S1 silencing with muscimol by saying "to confirm that S1 is required for this task..." Based on Hong and Bruno's experiments where transient silencing of S1 interrupted detection task performance, but lesioning S1 caused compensation elsewhere in the circuit and thus did not, it seems plausible to me that transient silencing with muscimol could interrupt the discrimination task because S1 is used in the task, but maybe not required under all circumstances. Therefore might it be safer to say that S1 is used? This may be a minor semantic issue.

We agree with the reviewer that a transient inhibition of S1 does not prove that S1 is required in all circumstances in our task. We changed the text accordingly.

7. In the Reporting Summary, the authors do use a Pearson correlation to assess whether the neurons encode whisking and licking, but do not discuss effect sizes, so this check box should not be N/A.

We apologize for the mistake, we now checked the box in the Reporting Summary file.

8. In the legend for Supplemental Fig. 2a (example of mRuby2 expression, DAPI staining, and GABA immuno) the text reads "Note that the vast majority of GABA-positive neurons is negative for GABA and vice-versa." This should read "the vast majority of mRuby2-positive neurons"?

Thanks for spotting this mistake. We fixed it.

Reviewer #2 (Remarks to the Author): Comments to the Authors: Chéreau, Bawa, Fodoulian, Carleton, Pagès and Holtmaat examines the effects of reward contingency reversal in mouse primary somatosensory cortex neurons using a Go/No-Go texture discrimination task. Mice were trained to the 'go/no-go' texture discrimination task and neuronal responses were monitored using GCaMP6s and 2PLSM. Upon finding a bias for P120 texture-selective neurons when pairing the P120 texture with reward, the group addressed whether this bias was due to an intrinsic bias toward coarser textures, or whether the bias is due to population encoding of higher order features, by switching the rewarded texture from P120 to P280. Ultimately, it was found that some neurons switched their selectivity to the P280 texture, implying that these neurons remained selective to the rewarded 'go' texture. While many neurons were considered to gain or lose a heightened discrimination response between textures, the authors did find some cells whose activity appears to encode the same information (value, identity, or choice) across contexts. Further analysis revealed that the response of value-selective neurons changes prior to performance improvements based on error history. It builds upon recent studies examining texture discrimination during learning, by asking whether those responses were stable across different contexts. The experiments and analysis are both solid and thorough. Overall, this investigation is very useful in that it elucidates possible features encoded by S1 that were initially thought to be exclusively 'higher order', including stimulus associated reward value or behavioral choice and it further suggests that an a priori preference for a specific texture in S1 may not exist. Major Points While there are several interesting findings reported in the manuscript, the main issue is clarity. The way it is presented is very confusing. I can understand the author's logic in starting with broad categorical breakdowns of functional types and sub-dividing these classes as the text proceeds, but I think this logic is distracting to the main points of the manuscript. Further, references to "selectivity", "go/no go", "behavior", "choice", "reward" are expressed in a way that is also confusing because their definition shifts through the course of the manuscript.

We thank the reviewer for her/his praise and useful comments. We have now revised the manuscript to improve the logic in the narrative.

1) The authors spend Figs. 1-3, examining P120-vs.P280-selectivity and Go/No-Go selectivity from the perspective of pre-reversal and then post-reversal learning and then re-analyzes these responses with respect to texture-selective, choice-selective, and value-selective neurons in S1 (Fig 4-5.). This is disorienting for the following reasons: a) P120-vs.P280-selectivity and Go/No-Go selectivity are dynamic definition which changes depending on pre- and post-reversal learning conditions and thus makes it difficult for the reader to keep track of what the author is referring to. b) P120-vs.P280-selectivity and Go/No-Go selectivity actually reflect some combination of texture, choice, and value selective neurons. After providing a sense for their distribution and breakdown in Figs. 2i, and 3d-h, it is not clear how these same neurons map onto the texture-selective, choice-selective, and value-selective neurons reported in the later figures? b) It makes more sense to define and report on texture-selective, choice-selective, and value-selective neurons (the main points) from the beginning of the manuscript. These are "static" definitions that don't change depending on the pre-reversal or post-reversal. The fraction of neurons for each of these functional cell classes and how they change over the course of reversal learning should then be presented. Shifting to P120-vs.P280-selectivity and Go/No-Go selectivity should be avoided as much as possible.

We agree with the reviewer that our presentation of the feature selectivity of neurons lacked some logic. Since Reviewer 1 provided feedback along similar lines, we have commented on this point in our 'general reply to the reviewers'. In the new version of the manuscript and figures we are now presenting all data in terms of texture selectivity, since this represents the most unambiguous form of selectivity that can be assessed from the beginning on, i.e. even before the training has started (e.g. Fig. 2). We appreciate the reviewer's suggestion to define all classes from the beginning on. However, we felt that this would greatly complicate the narrative. The reason for this being that further categorization of the subclasses could only be done after reversal learning. Only after reversal could we identify those neurons that remained selective for the original texture (termed 'texture neurons'), those that gained or lost selectivity ('gained' and 'lost neurons'), and those that reversed their selectivity to the other texture (which we formerly termed Go/No-go neurons). By using the Choice Index (CI) the latter class could be subdivided into neurons that were encoding the texture cue strictly based on upcoming choice to lick ('choice neurons') or on the value that was attached to it ('value neurons'). In the new manuscript we have now followed this logic throughout the text and figures, and adjusted the color scheme accordingly, i.e. dark brown for P120 selectivity and light brown for P280 selectivity. And then upon reversal, blue for neurons that gained selectivity, green for neurons that lost selectivity, red for those that reversed selectivity, and purple or orange for the 'choice' and 'value' neurons, respectively.

2) Similarly, the authors need to be more careful in how they discuss licking, whisking, and choice as it relates to behavior. The word behavior is used multiple times and very loosely applied to different things in a manner that also makes things confusing. Terms should be more clearly and conservatively defined. For example, there is a section on "L2/3 neurons selectivity report sensory input and not behavioral output". Behavioral output constitutes many things. Only observed motor parameters that include licking and whisking have been analyzed while other motor parameters have not. The authors suggest that "licking behavior" is not encoded in S1 but "choice behavior" is. Yet, licking in this task is the motor behavior that expresses the animal's choice. In fact, what the authors are describing are paired stimulus associations. From this standpoint "texture-selective" neurons can be referred to as "stimulus-only", "choice selective" can be referred to as "stimulus-response" neurons and "value selective" can be referred to as "stimulus-outcome" neurons.

We agree with the reviewer that some of the terms were used too loosely. We have thoroughly revised the text and figures, trying to be more conservative and precise in using certain definitions. As the reviewer points out, the experiments clearly reveal that neurons in S1 represent stimulus associations and not merely single or lower-order sensory features. We agree with her/his assessment, and carefully went through the results and discussion sections to make sure that this message is being conveyed. We prefer to stick to our original terms of 'choice' and 'value' neurons as this allowed us to reserve the term 'outcome' for the 'hits, misses, CRs, and FAs', and the term 'response' for 'neuronal activity/calcium signals'. Though, we did adapt the reviewer's explanation of this paired stimulus association, and incorporated it in the results and discussion section.

Minor Points:

- There are a few sweeping statements that don't seem to have a clear basis for support. For example, "even though a preference is probably absent at the population level for the textures that we used" We have changed this statement.
- A note on how long it took to reach expert criterion after reversal would have been appreciated

We have added this information to the main text: “Upon reversal the performance initially dropped to chance level (post-reversal naïve; Fig. 3b) before it reached the expert criterion again **within 2-4 days** (post-reversal expert).”

- The caption of Supplementary Figure 2 (“Note that the vast majority of GABA-positive neurons...”) is not worded clearly.

This has been fixed.

Reviewers' Comments:

Reviewer #1:

Remarks to the Author:

Data presentation are clarity are in general much improved.

The authors should Boldface the specific rewarded texture for the key in each figure (this is already done in Fig 4c, but it could be done throughout; for example in Figure 5a)

Terminology is still confusing: see for example Go cue and No-go cue. There is a cue for both types of trials (and indeed it appears to be the same cue). What does this mean? Am I mistaken that the cue is the same for both textures?

"Among those, a subpopulation of neurons regained selectivity contingent on stimulus-value."

This sentence is particularly hard to read. They are selective, then they become non-selective, then they become selective again? Isn't there a stimulus value for all trials – because there is always one of the textures that is rewarded.

The manuscript is still bogged down by jargon, the most critical of which is the definition of a value-sensitive neuron. This should be carefully defined in the introduction and then in the results there should be an elaborated description of how a neuron would be allocated to the "value-sensitive" group. Specifically, is it correct to say that there are neurons where a retrospective analysis indicates that their activity is higher in trials where there is an upcoming reward, even prior to the reward (this is a little spooky, so a careful description of how these cells are parceled out will be helpful).

"These value-sensitive neurons forecasted the onset of learning by displaying a distinct and transient increase in activity, depending on past behavioral experience." This effect is interesting and mechanistically provocative. Can the authors please speculate about what cellular and synaptic mechanisms might enable this, taking into account the specific structure of the trials (i.e., how long does the "trace" of the prior trial have to last, for the next trial to interact with it?).

Figure 5d and 4f,g show non-significant data – the statistical evaluation of these data are not really the point, because the tests used are not sensitive to small mean differences in a diverse population (i.e. with a large range). Perhaps they can be moved to a supplemental figure? It is distracting and also uninformative (since in many cases the bar graphs suggest that there is something different but the coarse statistical analysis did not validate this). It would be hard to convince the reader that these values are the same – instead, we are left with hanging without the blessing of some statistical function. The authors should come to grips with how they want to talk about something that they can't say is "the same" but that their statistical test won't validate as "different." Maybe explain why this is the case? How underpowered are they to detect a potential difference if there is one?

Reviewer #2:

Remarks to the Author:

The authors have addressed all my concerns. This is a great contribution to the field.

Jerry Chen

Response to Reviewers' comments: (response to reviewers in blue)

Reviewer #1 (Remarks to the Author)

Data presentation are clarity are in general much improved.

We thank the reviewer for helping us to improve the manuscript.

The authors should Boldface the specific rewarded texture for the key in each figure (this is already done in Fig 4c, but it could be done throughout; for example in Figure 5a)

We followed the reviewer's suggestion and boldfaced the rewarded texture throughout.

Terminology is still confusing: see for example Go cue and No-go cue. There is a cue for both types of trials (and indeed it appears to be the same cue). What does this mean? Am I mistaken that the cue is the same for both textures?

We agree that the term "cue" for the rewarded (Go-cue) or non-rewarded texture (No go-cue) could be confusing because it also refers to the sound that precedes the texture presentation. We have now replaced the term to "rewarded" and "non-rewarded textures" where appropriate.

"Among those, a subpopulation of neurons regained selectivity contingent on stimulus-value."
This sentence is particularly hard to read. They are selective, then they become non-selective, then they become selective again? Isn't there a stimulus value for all trials – because there is always one of the textures that is rewarded.

We agree with the reviewer that this particular sentence in the abstract was not very clear. We changed it to convey better that those neurons regain selectivity for a texture contingent on the associated reward value: "... but the majority was dynamic. Among those, a subpopulation of neurons regains texture selectivity contingent on the associated reward value." It is important to note that the cells do not report the reward as such, they reshape their texture selectivity congruent with the reward that is associated with it during the learning phases of the task.

The manuscript is still bogged down by jargon, the most critical of which is the definition of a value-sensitive neuron. This should be carefully defined in the introduction and then in the results there should be an elaborated description of how a neuron would be allocated to the "value-sensitive" group. Specifically, is it correct to say that there are neurons where a retrospective analysis indicates that their activity is higher in trials where there is an upcoming reward, even prior to the reward (this is a little spooky, so a careful description of how these cells are parceled out will be helpful).

We have extended the introduction to pave the way for the term value-sensitive neurons, and now highlight the result that those neurons first lose and then regain texture selectivity contingent on the associated reward upon reversal learning: "We then reassess their selectivity upon reversal learning, which reveals a substantial subset of neurons that dynamically represents textures. Many lose or gain selectivity. Yet another class, which we term value-sensitive neurons, first lose and then regain texture

selectivity contingent on the associated reward. The ramping up of this selectivity forecasts the onset of learning.”

With respect to the second part of the reviewer’s comment, our retrospective analysis shows that neurons increasingly respond to the texture that is associated with the reward. It would be too speculative to say that they predict the reward. We can only go as far as stating that they increase their sensitivity to a texture when this texture is repeatedly followed by a reward during training. We have carefully re-read the discussion on this point and believe that this is specified.

“These value-sensitive neurons forecasted the onset of learning by displaying a distinct and transient increase in activity, depending on past behavioral experience.” This effect is interesting and mechanistically provocative. Can the authors please speculate about what cellular and synaptic mechanisms might enable this, taking into account the specific structure of the trials (i.e., how long does the “trace” of the prior trial have to last, for the next trial to interact with it?).

This is an interesting point. Indeed, one could speculate that the rewards or absence thereof evokes a long-lasting effect that modifies the response in the upcoming trial. We cannot assess or measure this specifically since our recordings stopped at maximally 1 sec after the texture presentation and we did not record between trials. That said, a visual inspection of the traces suggests that neuronal responses in FA trials did not last longer than those in Hit trials. We favor the explanation that value-sensitive neurons are “tuned” by modulation, which occurs as a result of prior behavioral outcomes. We elaborate on this in two parts of the discussion:

Page 11. “Previous studies indicate that the value of a sensory stimulus is encoded by higher-order areas such as the posterior parietal, orbitofrontal, and retrosplenial cortices^{8, 9, 26, 27}. Our data shows that value-encoding is also an attribute of a population of neurons in S1. The instructive cues for this selectivity could be manifold. For example, they could be provided by direct feedback from the aforementioned higher-order cortical areas, or they could be derived from sub-cortical areas that are implicated in attention and behavioral updating during learning^{32, 33}. Modulatory reinforcement signals that are associated with behavioral outcome could also play a major role^{33, 34, 35}. Indeed, reward-related response modulation has been observed in S1²⁸, and was found to promote cortical plasticity processes related to visual response tuning in primary visual cortex¹⁶.”

Page 12. “During the naïve reversal phase these neurons showed higher responses in hit trials if the hit trial was preceded by a false alarm trial. This phenomenon was prominent during the transition from the naïve to expert reversal phase and forecasted the increase in behavioral performance. We speculate that the omission of reward-associated signals during a false alarm trial directs the animal’s attention towards the newly rewarded texture. Elevated attentional signals have been shown to modulate sensory-driven responses in visual cortex⁴².”

Figure 5d and 4f,g show non-significant data – the statistical evaluation of these data are not really the point, because the tests used are not sensitive to small mean differences in a diverse population (i.e. with a large range). Perhaps they can be moved to a supplemental figure? It is distracting and also uninformative (since in many cases the bar graphs suggest that there is something different but the coarse statistical analysis did not validate this). It would be hard to convince the reader that these values are the same – instead, we are left with hanging without the blessing of some statistical function. The authors should come to grips with how they want to talk about something that they can’t say is “the

same” but that their statistical test won’t validate as “different.” Maybe explain why this is the case? How underpowered are they to detect a potential difference if there is one?

We believe that the confusion may have been caused by the fact that we labelled the graph with “all n.s.”. This may have provoked the impression that we compared the responses between classes. However, in these panels (Figure 4fg and 5d), we only compared the calcium signals between low vs high whisking or licking rate trials within each class and not across classes (as specified in the figure legends). The absence of significant differences between high and low whisking or licking rate trials does not prove that there is no difference (as the reviewer notes) but at least takes away the possibility that the inferred selectivity was merely the result of differences in behavior. Therefore, we think that it is important to keep this information in the main figure. Nonetheless, in order to improve the readability of the graphs, we now represent the data with connected dots instead of individual bars. The associated box plots are now presented in supplementary figure 4, which provides a better appreciation of the variance in the calcium signals between the classes.

Reviewer #2 (Remarks to the Author)

The authors have addressed all my concerns. This is a great contribution to the field.

We thank the reviewer for helping us to improve the manuscript.